# Learn to Change the World: Multi-level Reinforcement Learning with Model-Changing Actions

## Abstract

Reinforcement learning usually assumes a given or sometimes even fixed environment in which an agent seeks an optimal policy to maximize its long-term discounted reward. In contrast, we consider agents that are not limited to passive adaptations: they instead have model-changing actions that actively modify the RL model of world dynamics itself. Reconfiguring the underlying transition processes can potentially increase the agents' rewards. Motivated by this setting, we introduce the multi-layer configurable time-varying Markov decision process (MCTVMDP). In an MCTVMDP, the lower-level MDP has a non-stationary transition function that is configurable through upper-level model-changing actions. The agent's objective consists of two parts: Optimize the configuration policies in the upper-level MDP and optimize the primitive action policies in the lower-level MDP to jointly improve its expected long-term reward.

## 1 Introduction

Reinforcement learning, which is based on the mathematical model Markov decision process (MDP), has been widely applied in many real-world sequential decision problems, such as in RLHF Christiano et al. (2023), in financial decisions Liang et al. (2018), and in robotic control algorithms Tang et al. (2024), etc. There are extensive works in finding optimal policies in a fixed environment, through both model-based approaches Deisenroth & Rasmussen (2011) such as planning on estimated models, and model-free approaches such as variants of Q-learning Watkins & Dayan (1992) and variants of policy gradient methods Williams (1992). However, in many applications, an MDP environment can be changed on purpose in order to increase the obtainable rewards.

In this paper, we consider a new RL framework, where an agent has actions to change the involved MDP environment itself. These changes include, but are not limited to, changing its transition kernels, rewards, and even the set of allowable future actions. The traditional RL framework behaves within a pre-specified statistical model, and this statistical model stays unchanged no matter what actions are taken by the agents. In contrast, the new RL framework proposed in this paper has the potential and mechanism of breaking out of these limitations, through actions that change or improve the underlying MDP. For example, RL agents may have actions that can change the transition kernels. After such a model-changing action is taken at a certain times step, the transition kernel is updated and stays unchanged until the next model-changing action occurs.

As special cases of the proposed RL mechanism with model-changing actions, we consider two schemes: **multi-level (including bi-level) environment-changing RL** and **configurable RL for time-varying environments**. In configurable RL for a time-varying environment, after a certain number of time steps, the transition kernel is changed to a different transition kernel by nature. At this time step, the agent takes action that can change or improve the transition kernel. In the bi-level environment-changing RL scheme, in the lower-level MDP, the agent aims to find the regular optimal policy given a certain environment configuration, while in the upper-level MDP, the agent aims to find the optimal configuration policy for modifying the lower-level MDP. The lower-level MDP's quantities, which the upper-level MDP can configure, thus become the states of the upper-level MDP. For example, if the upper-level MDP explores actions that can improve the lower-level MDP's transition kernel, the lower level's transition kernel becomes the state of the upper-level

MDP. In a regular MDP model, the states and the transition kernel are completely different subjects or concepts; however, in the unique multi-level MDP setting considered in this paper, the lower-level transition kernel can become the state of the upper-level MDP. To the best of our knowledge, the formulations that treat the transition kernels themselves as upper-level MDP states are rare or have not previously appeared in the literature.

Let us consider an example of a 3-level MDP framework with model-changing actions. The highest level, namely the 3rd level, of this framework is an MDP which represents the evolution of the politics and legislation of a country: this MDP's states are the evolving approved guidelines (fiscal policies) for this country's central bank, and the actions are the efforts of legislation. Note that legislative efforts can potentially result in random fiscal policies that depend on unexpected political compromises and random political events. The 2nd-level MDP represents the dynamics of the central bank setting up and exploring monetary policies: the states of this 2nd-level MDP are the monetary policies, and the transition kernel between the monetary policies is affected by the fiscal policies set by the 3rd-level MDP. The actions on this 2nd level are monetary policy explorations, such as motions to change the Federal Reserve interest rates. These actions may result in random-sized interest rate changes due to monetary policy voting results and random foreign countries' economic environments. The 1st-level MDP represents the society's economic activities: this lower-level MDP's states are the situations of the society's productions and consumptions of goods and services, and the transition kernel between the states is dictated or configured by the monetary policies adopted in the 2nd-level MDP.

Consider another example of a robot trying to cross a fast-flowing river. Without changing the environment, the robot performs RL on non-model-changing maneuvers e.g., moving left, right, backward, forward, upward, or downward) to adapt to the river flow, but may still be swept away with a high probability if the current is too strong. In our model-changing RL setting, the robot can instead modify the environment of this river, for example, by placing stepping stones in the river. This naturally leads to a bi-level MDP with model-changing actions. The state in the upper-level MDP is the configuration of the river environment, dictating the transition kernel for the lower MDP for the robot's non-model-changing maneuvers. The upper-level MDP's actions are the robot's actions, which change the configuration of the river environment. We note that the robot's actions of putting stepping stones into the river can lead to transitions to random configurations of the river environment, because it is random whether a deployed stone is washed away or stays still in place under the river current. The lower-level actions are the robot's non-model-changing maneuvers under a given river environment. The states affected by the lower-level actions are the locations of the robot in the river. Please refer to Appendix A for more motivating examples on transportation on infrastructure, training drones, and finance models.

Motivated by these examples, this paper makes the following contributions: 1) We formulated the problem of RL with model-changing actions; and we proposed two special models for RL with model-changing actions: multi-level configurable MDPs and time-varying configurable MDP; 2) We proposed algorithms including convex optimization formulations and multi-level value iterations for solving multi-level configurable RL problems; 3) We proved theoretical performance guarantees for the proposed algorithms; 4) We provided numerical results showing the effectiveness of configuring or improving favorable RL environment through learning.

**Related works**: The literature most closely related to our work is *configurable MDP* (CMDP) Metelli et al. (2018); Silva et al. (2019); Chen et al. (2022); Thoma et al. (2024); Silva et al. (2018); Modhe et al. (2021); Maran et al.; Ramponi et al. (2021), where the agent can configure some environmental parameters to improve the performance of a learning agent. Within the series of works in configurable MDP, Silva et al. (2018) assumes that a better world configuration corresponds to a transition probability matrix whose corresponding optimal policy yields a larger total discounted reward. They formulate the problem of reasoning over good world configurations as a non-convex constrained optimization problem, where the agent explicitly balances the benefits of changing the world against the costs incurred by such modifications. Unlike our approach, their formulation does not employ upper-level MDP abstractions/learning to model/improve configuration actions, but relies on direct gradient-based optimization. In addition, our paper deals with time-varying non-stationary lower-level MDPs.

Thoma et al. (2024) addresses how to optimize configurations for a contextual MDP where some parameters are configurable while others are stochastic. Their bi-level gradient-based formulation

(BO-MDP) can be viewed as a Stackelberg game where the leader and a random context beyond the leader's control together configure an MDP, while (potentially many) followers optimize their strategies given the setting. Chen et al. (2022) focuses on regulating the agent's interaction with the environment by redesigning the reward or transition kernel parameters. They formulate this problem as a bi-level program, in which the upper-level designer regulates the lower-level MDP, aiming to induce desired policies in the agent and achieve system-level objectives. In contrast with these works, our work places the configuration process under the control of the RL agent itself, which adjusts/configures the lower-level kernels with the goal of maximizing its own reward. In our work, the RL model has built-in actions that can change the model itself, which is not the case in previous works. Moreover, in our setting, upper-level configurations are explicitly modeled as an MDP, which is unlike the one-time configuration in previous works. Silva et al. (2019) analyzes the complexity of solving CMDPs, demonstrates several parameterizations of CMDPs, and derives a gradient-based solution approach. Their approach, particularly in the continuous configuration setting, is related to our linear approximation method for solving the special case of TVCMDPs. However, our primary contribution lies in the study of multi-layer configurable MDPs. which differs fundamentally from the continuous framework in Silva et al. (2019). Our work also covers time-varying transition kernels, which were not investigated by these previous works. A unique characteristic of our framework is that the transition kernels of lower-level MDPs actually become the states of an upper-level MDP. For further comparisons with hierarchical MDP Li et al. (2022), meta RL Duan et al. (2016), and semi-MDP Sutton et al. (1999), please see Appendix B.

## 2  PROBLEM FORMULATION

**General idea of a framework with model-changing actions**: We consider MDP with model-changing actions $\mathcal{M}_C = \{\mathcal{S}, \mathcal{A}, P, T, r, \gamma\}$ with model-changing actions that can modify its transition kernel. In $\mathcal{M}_C$, $\mathcal{S}$ is the state space with $|\mathcal{S}| = n$, $\mathcal{A}$ is the action space, the transition kernel $P : \mathcal{S} \times \mathcal{A} \times \mathcal{S} \to [0, 1]$ is subject to change or configuration and may be time-varying. $r$ is the reward function, $T$ is the time steps, and $\gamma$ is the discounting factor. Different from a regular MDP, at certain time steps, the agent may adopt model-changing actions which can change the MDP to have a new (maybe random) transition kernel, and this new transition kernel will remain fixed until another new model-changing action adopted in the future changes the transition kernel again.

Because the transition kernels can change over time due to model-changing actions, most traditional theories for MDP do not apply. We consider the following special cases called multi-level configurable MDP where the time steps are divided into episodes, and the model-changing actions are only taken at the beginning of each episode. The model-changing actions are actions of upper-level MDPs which dictate or change the transition kernels of lower-level MDPs. We focus on bi-level configurable MDPs, which can be extended to multi-level MDP in a similar way.

**Bi-level configurable MDP**: We build a bi-level configurable MDP that separates model-changing actions (configuration operations) from primitive actions, using an episodic-style hierarchical structure. For the **upper-level** model, the agent chooses a model-changing action at the start of each episode that configures the lower-level environment by selecting a lower-level transition kernel. This configuration determines the dynamics for the entire episode. Sequential decisions on model-changing actions throughout episodes form an upper-level MDP, aiming to optimize the model-changing policy and to improve the lower-level model over time. There is a cost imposed on taking a model-changing action, which is represented as a penalty included in the upper-level reward function. For the **lower-level** model, within each episode, the agent interacts with a standard MDP with the current transition kernel set by the upper level. The agent normally selects primitive actions, receives rewards, and aims to optimize its policy. We use $k$ to denote the episode index and use $t$ to denote the time step index within an episode.

Mathematically, the bi-level MDP can be formulated as: a lower-level $\mathcal{M}_L = \{\mathcal{S}, \mathcal{A}, P, T, \mu_0, r, \gamma\}$ and an upper-level $\mathcal{M}_U = \{\mathcal{P}, \mathcal{B}, Q, K, R, \lambda\}$. In $\mathcal{M}_L$, which can be considered similar to a standard MDP, $\mathcal{S}$ is the state space with $|\mathcal{S}| = n$, $\mathcal{A}$ is the action space, the lower-level transition kernel $P : \mathcal{S} \times \mathcal{A} \times \mathcal{S} \to [0, 1]$ is determined by $\mathcal{M}_U$ and is subject to change. We denote the transition kernel in episode $k$ as $P_k$. $T$ is the number of time steps within one episode and can be infinity, when we consider an infinite-horizon $\mathcal{M}_L$. $\mu_0 \in \mathbb{R}^n$ is the initial state distribution of each episode and is set to be uniform. At the beginning of each episode $k$, the state distribution is reset

to be $\mu_0$. $r : \mathcal{S} \times \mathcal{A} \rightarrow \mathbb{R}$ is the reward function and we denote $r_{\max} = \max_{(s,a) \in \mathcal{S} \times \mathcal{A}} r(s, a)$. $\gamma \in [0, 1)$ is the lower-level discount factor.

We use $\pi_k : \mathcal{S} \rightarrow \mathcal{A}$ to denote the primitive policy of episode $k$ that determines the lower-level actions of the agents. For episode $k$, the lower-level state-value (namely V-value) function of the agent following the policy $\pi_k$ is:

$$V^{\pi_k}(s) = \mathbb{E}_{P_k}[\sum_{t=0}^{\infty} \gamma^t r(s_t, a_t) | s_0 = s, a_t \sim \pi_k], s \in \mathcal{S}$$

and the closed-form solution of the Bellman equation for the state-value function is:

$$V^{\pi_k} = (I - \gamma P_k^{\pi_k})^{-1} r^{\pi_k}, \tag{1}$$

where $V^{\pi_k, P_k} \in \mathbb{R}^n$ is the vectorized state-values, and $P_k^{\pi_k}, r^{\pi_k}$ are in the form:

$$P_k^{\pi_k} = \begin{bmatrix} -P_k(\cdot|s_1, \pi_k(s_1))- \\ -P_k(\cdot|s_2, \pi_k(s_2))- \\ \vdots \\ -P_k(\cdot|s_n, \pi_k(s_n))- \end{bmatrix}, r^{\pi_k} = \begin{bmatrix} r(s_1, \pi_k(s_1)) \\ r(s_2, \pi_k(s_2)) \\ \vdots \\ r(s_n, \pi_k(s_n)) \end{bmatrix}.$$

We denote the lower-level initial expected return $J(\pi_k, P_k)$ of episode $k$ as:

$$J(\pi_k, P_k) = \mu_0^T V^{\pi_k, P_k}. \tag{2}$$

In $\mathcal{M}_U$, $\mathcal{P}$ is the space of lower-level transition kernels and is the state space of $\mathcal{M}_U$. For simplicity, we assume that the upper-level state $\mathcal{P}$ with $|\mathcal{P}| = m$ is discrete and finite. The upper-level state $P \in \mathcal{P}$ is also the transition kernel $P$ of the lower-level MDP $\mathcal{M}_L$. $\mathcal{B}$ is the set of model-changing actions. $Q : \mathcal{P} \times \mathcal{B} \times \mathcal{P} \rightarrow [0, 1]$ is the upper-level kernel that determines the transitions of upper-level states. The notation of the upper-level kernel is $Q(P'|P, b)$ where $P'$ is the configured lower-level kernel, and the distribution of $P'$ depends on the current lower-level kernel $P$ and the model-changing action $b$. $K$ is the total number of episodes and can go to infinity when we consider an infinite-horizon $\mathcal{M}_U$. $\lambda \in [0, 1)$ is the upper-level discount factor. $R : \mathcal{P} \times \mathcal{B} \rightarrow \mathbb{R}$ is the upper-level reward function. The reward of episode $k$ is defined as:

$$R(P_k, b_k) = \sum_{P_{k+1} \in \mathcal{P}} Q(P_{k+1}|P_k, b_k) J(\pi_{k+1}^*, P_{k+1}) - C(P_k, b_k), \tag{3}$$

where $\pi_{k+1}^*$ is the optimal primitive policy of episode $k + 1$, and $C(P_k, b_k)$ is the cost function. We denote $R_{\max} = \max_{(P,b) \in \mathcal{P} \times \mathcal{B}} R(P, b)$.

We let $W^\Theta(P)$ denote the higher-order state-value function if the agent follows the higher-order policy of configuration operations $\Theta : P \rightarrow \mathcal{B}$:

$$W^\Theta(P) = \mathbb{E}_Q[\sum_{k=0}^{\infty} \lambda^k R(P_k, b_k) | P_0 = P, b_k \sim \Theta], P \in \mathcal{P}$$

The agent aims to find an optimal higher-order policy $\Theta^*$ such that $\Theta^* = \arg\max_\Theta W^\Theta$.

**Special case of bi-level configurable MDPs: Time-variant MDP and continuous configuration**: We consider a special case when the upper-level state space $\mathcal{P}$ is continuous and agents can continuously change the lower-level environment deterministically. In this case, we only consider a one-layer MDP, in which the transition kernel is time-variant throughout episodes and configurable. Additionally, the configuration operations incur costs when the agent modifies a less favorable transition kernel to a more favorable one. We study a constrained optimization problem that seeks to maximize the agent's discounted long-term reward asymptotically as time goes to infinity, taking into account both the time-varying world dynamics and a budget constraint on the total cost of configurations.

Mathematically, the time-variant configurable MDP (TVCMDP) can be described as the following: $\mathcal{M}_{TVC} = \{\mathcal{S}, \mathcal{A}, \mathcal{C}, K, \{P_k\}_{k=1}^K, \mu_0, T, r, \gamma\}$, where $\mathcal{C}$ is the space of configuration operations, $K$ is the total number of episodes, and $T$ is the total number of time steps within each episode. The

time-varying transition kernel $P_k, k \in [K]$ is determined by nature at the beginning of each episode (for example, without maintenance, road can randomly deteriorate after a period of time). All other parameters are the same as those introduced in $\mathcal{M}_L$.

Within episode $k$, the configuration operation $x \in \mathcal{C}$ is represented by $x_k \in [-1,1]^{n \times n}$ where $x_k$ is the amount of change the agent does to the default transition kernel $P_k^{\pi_k}$. To evaluate the effect of configurations $\mathbf{x} = \{x_k\}_{k=0}^K$, we define the sum of configured function $F(\mathbf{x}; \boldsymbol{\pi})$ with the set of configuration operations $\mathbf{x} = \{x_k\}_{k=0}^K$ and the set of primitive policies $\boldsymbol{\pi} = \{\pi_k\}_{k=0}^K$ as:

$$F(\mathbf{x}; \boldsymbol{\pi}) = \sum_{k=0}^K J(\pi_k, P_k^{\pi_k} + x_k), \tag{4}$$

where $J(\pi, P)$ is determined by (2). The agent in TVCMDP aims to maximize the objective function (4) by optimizing the configuration variables $\mathbf{x}$.

## 3 COST-CONSTRAINED OPTIMIZATION PROBLEM ON TVCMDP

**Cost constrained optimization problem**: Recall that in TVCMDP, the agent wants to maximize the objective function (4) by optimizing the configuration variables $x_k$ throughout the $K$ episodes. The original cost-constrained objective function under the configuration budget is (5),

$$\max_{x_k} \max_{\pi_k} \sum_{k=0}^K J(\pi_k, P_k^{\pi_k} + x_k), \qquad (5) \qquad \max_{x_k} \sum_{k=0}^K \langle A_k, x_k \rangle, \tag{6}$$

$$\text{s.t.} \sum_{k=0}^K C(x_k) \le B, \qquad\qquad\qquad \text{s.t.} \sum_{k,i,j} \left(e^{\alpha|(x_k)_{ij}|} - 1\right) \le B,$$

$$\sum_{j=1}^n \left(P_k^{\pi_k} + x_k\right)_{ij} = 1, \quad \forall i, k, \qquad \sum_{j=1}^n (x_k)_{ij} = 0, \quad \forall i, k,$$

$$0 \le \left(P_k^{\pi_k} + x_k\right)_{ij} \le 1, \quad \forall i, j, k, \qquad 0 \le \left(P_k^{\pi_k} + x_k\right)_{ij} \le 1. \quad \forall i, j, k,$$

where $B$ is the configuration budget, $\pi_k$ is the policy in episode $k$, $P_k$ is the original transition kernel in episode $k$, and $C(x_k)$ is the cost function of changing the default transition kernel by $x_k$. Because configuration to the environment may be a highly-costly operation, we assume that the cost grows exponentially as the amount of configuration increases, i.e., $C(x) = \sum_{ij} \beta(e^{\alpha|(x)_{i,j}|} - 1)$, where $\alpha, \beta \in \mathbb{R}$ are non-negative constants, and $\alpha$ can be large. To solve this cost-constrained optimization problem, we first linearize the objective function (4). As a by-product, we also solve out the Jacobian $\nabla_{P^\pi} V^\pi \in \mathbb{R}^{n \times n \times n}$ of $V^\pi$ with respect to $P^\pi$. Please see Appendix C.

**Linear approximation of configured state-values**: Consider the closed form solution in (1), we have that in any episode $k$, for a fixed policy $\pi$, the state-value is $V^\pi = (I - \gamma P^\pi)^{-1} r^\pi, V^\pi \in \mathbb{R}^n, P^\pi \in [0,1]^{n \times n}, r^\pi \in \mathbb{R}^n$. Suppose the optimal policy corresponding to the original transition kernel $P$ is $\pi$. We assume that with a sufficiently small change $x \in [-1,1]^{n \times n}$ in the transition kernel, the optimal policy $\pi$ remains unchanged. Even if the optimal policy changes when we change $P$ to a different transition kernel $P_{new}$, we notice that $V^\pi(P_{new}) = (I - \gamma P_{new}^\pi)^{-1} r^\pi$ is still a lower bound on the state-value of the optimal policy under the new transition kernel $P_{new}$. This is because $\pi$ is the optimal policy under $P$, but not necessarily optimal under $P_{new}$. So in our optimization, keeping the policy $\pi$ unchanged while $P$ changes provides a meaningful lower bound for the state-value.

The configured state-value function $V^\pi(P^\pi + x)$ by linear approximation is computed by:

$$V^\pi(P^\pi + x) = \left(I - \gamma(P^\pi + x)\right)^{-1} r^\pi$$
$$\approx (I - \gamma P^\pi)^{-1} r^\pi - (I - \gamma P^\pi)^{-1}(-\gamma x)(I - \gamma P^\pi)^{-1} r^\pi$$
$$= (I - \gamma P^\pi)^{-1} r^\pi + \gamma(I - \gamma P^\pi)^{-1} x (I - \gamma P^\pi)^{-1} r^\pi$$

We let matrix $M^\pi = \gamma(I - \gamma P^\pi)^{-1}$, $M^\pi \in \mathbb{R}^{n \times n}$, and let vector $N^\pi = (I - \gamma P^\pi)^{-1} r^\pi$, $N^\pi \in \mathbb{R}^n$. The above formula can be rewritten as:

$$V^\pi(P^\pi + \mathrm{x}) \approx N^\pi + M^\pi \mathrm{x} N^\pi \tag{7}$$

**Convex optimization problem**: We can now rewrite the original optimization problem (5) by applying the linear approximation (7). Now let $\pi_k$ be the optimal policy under the original transition kernel $P_k$ in episode $k$. Now $N_k^{\pi_k} \in \mathbb{R}^n$ and $M_k^{\pi_k} \in \mathbb{R}^{n \times n}$ are associated with episode $k$. The objective function can be rewritten as $\max_{\mathrm{x}_k} \sum_{k=0}^K \mu_0^T N_k^{\pi_k} + \mu_0^T M_k^{\pi_k} \mathrm{x}_k N_k^{\pi_k}$. We can ignore the constant term $\mu_0^T N_k^{\pi_k}$ with respect to x, because we expand the objective function at $\pi_k$, which is just a function of the original transition kernel $P_k$ and does not change with respect to the configuration variable x. Since $\mu_0^T M_k^{\pi_k} \mathrm{x}_k N_k^{\pi_k}$ is a scalar, we have $\mu_0^T M_k^{\pi_k} \mathrm{x}_k N_k^{\pi_k} = tr(\mu^T M_k^{\pi_k} \mathrm{x}_k N_k^{\pi_k})$, and $tr(\cdot)$ is the matrix trace function. According to the cyclic property of trace, we have $tr(\mu_0^T M_k^{\pi_k} \mathrm{x}_k N_k^{\pi_k}) = tr(N_k^{\pi_k} \mu_0^T M_k^{\pi_k} \mathrm{x}_k)$. Consider the property that $tr(A^T, \mathrm{x}) = \langle A, \mathrm{x} \rangle$, where $\langle \cdot, \cdot \rangle$ is the Frobenius norm. If we let $A_k^T = N_k^{\pi_k} \mu_0^T M_k^{\pi_k}$, this objective function is equal to $\langle A_k, x_k \rangle = \sum_{ij} (A_k)_{ij} (\mathrm{x}_k)_{ij}$, and $A_k = (M_k^{\pi_k})^T \mu_0 (N_k^{\pi_k})^T$, $A_k \in \mathbb{R}^{n \times n}$.

We assume that the cost function is point-wise exponential and $B$ is the total cost budget, i.e., $\sum_{k \le K, i,j \le n} (e^{\alpha|(\mathrm{x}_k)_{ij}|} - 1) \le B$, and the constant $\alpha$ is large. In order to make $P_k^{\pi_k} + \mathrm{x}_k$ a valid probability transition matrix, $\mathrm{x}_k$ also need to satisfy that $\sum_j (\mathrm{x}_k)_{ij} = 0, \forall i, k$, and $P_k^{\pi_k} + x_k$ has elements between 0 and 1. By reorganizing these constraints, we get a convex optimization problem with a linear objective function (6). Here, the optimization variables are the configurable variables $\{\mathrm{x}_k\}_{k=0}^K$. $\{\pi_k\}$ are the set of optimal policies the agent can adopt in episode $k$. This problem is solvable using classic convex optimization methods. Note that once we get the updated $P$, we can update the optimal policy under the new transition kernel. One can even redo this with another linear approximation at the original kernel under the new updated policy. Compared with previous works on configurable RL for fixed kernel Silva et al. (2018), our novelty for this special case is that we are dealing with time-varying non-stationary RL.

## 4 BI-LEVEL MODEL-BASED VALUE ITERATION

**Algorithm**: We propose the model-based Bi-level value iteration algorithm to solve the Bi-level configurable MDP model proposed in section 2. Please see Appendix D, which contains the estimation approach to get empirical transition kernels $\{\hat{P}\}$ and $\hat{Q}$ and the algorithm pseudo-code. To apply Algorithm 1, we need to first estimate the ground-truth lower-level kernels $P \in \mathcal{P}$ and the ground-truth upper-level kernel $Q$. We assume that the reward function $r(s, a), \forall (s, a) \in \mathcal{S} \times \mathcal{A}$ is known and remains the same across all possible lower-level models.

Algorithm 1 produces the higher-order state-values $W^{H_U}$ and the higher-order policy $\Theta^{H_U}$. The higher-order state-value represents the maximum achievable expected returns when the agent jointly optimizes both the environment configuration and its adaptation to the configured environment. The policy $\Theta^{H_U}$ specifies the optimal model-changing action, i.e., configuration, to apply to the current lower-level kernel. Each lower-level expected return $J$, which is the "average" of lower-level state-values, contributes to the upper-level reward function $R$ (based on equation (3)). The upper-level MDP makes decisions on model changing according to the information reported by the lower-level MDP. [1]

In the following section, we discuss how the physical limitation of the bi-level MDP model and the estimation error of the model-based algorithm would affect the performance of Algorithm 1. Here the physical limitation error comes from the precision limit of configuration, and the estimation error refers to error of lower-level MDP estimating its configured kernel.

**Physical limitation and estimation error**: Suppose that the upper-level MDP configures the lower-level MDP to the ideal lower-level kernel $P_c \in \mathcal{P}_c$. However, due to physical limitations on configu-

---

[1]We remark that for Algorithm 1, we assume that there are $m$ possible states for the upper-level MDP, namely there are $m$ possible transition kernels for the lower-level MDP. Due to configuration error or estimation error, the estimation $\hat{P}$ may not be exactly the same as one of the $m$ possible kernels; however, we assume that those errors are small such that we still regard $\hat{P}$ as "in" the set of $m$ candidate ideal transition kernels and know which set member $\hat{P}$ is closest to or associated with.

ration precision, the actual lower-level kernel is $P$, and the agent estimates it as $\hat{P}$. The discrepancy between the ideal $P_c \in \mathcal{P}_c$ and the empirical estimate $\hat{P} \in \hat{\mathcal{P}}$ can be decomposed into *physical discrepancy* (bounded by $\delta_c$) and *estimation error* (bounded by $\delta_g$). Both propagate upward, inducing state/reward perturbations in the upper-level MDP. Additionally, the upper-level estimation error between the upper-level true kernel $Q$ and its estimate $\hat{Q}$, bounded by $\Delta$, introduces another source of upper-level error.

*Physical limitation error (physical discrepancy)*: For any ideally configured transition kernel $P_c \in \mathcal{P}_c$, we assume that the true kernel $P \in \mathcal{P}$ lies within an uncertainty set centered around $P_c$. In particular, the uncertainty is imposed in a decoupled manner for each state-action pair $(s, a) \in \mathcal{S} \times \mathcal{A}$, satisfying the $(s, a)$-rectangularity condition Wiesemann et al. (2013): $\forall (s, a)$, the total variance distance between $P_c$ and $P$ is bounded by: $TV(P_c(\cdot|s, a), P(\cdot|s, a)) = 1/2\|P_c(\cdot|s, a) - P(\cdot|s, a)\|_1 \leq \delta_c$.

*Statistical error (estimation error)*: Due to the limited samples, we assume that the empirical kernel $\hat{P}$ lies within an uncertainty set centered around the true $P$. Specifically, $\forall (s, a)$, the total variance distance between $P$ and $\hat{P}$ is bounded by: $TV(P(\cdot|s, a), \hat{P}(\cdot|s, a)) \leq \delta_g$. Additionally, by applying the triangle inequality on the total variance distance, for $\forall (s, a) \in (\mathcal{S} \times \mathcal{A})$, the distance between $P_c$ and $\hat{P}$ is bounded by: $TV(P_c(\cdot|s, a), \hat{P}(\cdot|s, a)) \leq \delta_g + \delta_c$.

Similarly, for the ground-truth upper-level kernel $Q$ and the empirical $\hat{Q}$, the total variance distance is bounded by: $\forall (P, b) \in \mathcal{P} \times \mathcal{B}, TV(Q(\cdot|P, b), \hat{Q}(\cdot|P, b)) \leq \Delta$.

## 5 PERFORMANCE ANALYSIS

**Notations**: We define some extra notations for the following analyses: By executing the lower-level policy $\pi : \mathcal{S} \to \mathcal{A}$, and executing the higher-order policy $\Theta : \mathcal{P} \to \mathcal{B}$:

$V_{P_c}^\pi$: the **ideal** lower-level state-value of the ideally configured lower-level MDP with the ideal transition kernel $P_c$;

$V_P^\pi$: the **ground-truth** lower-level state-value of lower-level ground-truth kernel $P$ ($P$ may be different from $P_c$ due to configuration error);

$V_{\hat{P}}^\pi$: the **empirical** lower-level state-value of lower-level MDP with empirical transition kernel $\hat{P}$;

$W_Q^\Theta$: the **ideal** higher-order state-value of the upper-level MDP with ground-truth kernel $Q$, where we also assume the lower-level MDP has the ideal transition kernel $P_c$;

$W_{\hat{Q}}^\Theta$: the **empirical** higher-order state-value of the empirical upper-level MDP.

We now present the following estimation error lemma which characterizes the effect of estimation error, highlighting the performance gap which arises from the difference between the true transition kernel $P$ and its estimation $\hat{P}$.

**Lemma 1 (estimation error Lemma)** *If $\forall (s, a) \in \mathcal{S} \times \mathcal{A}$, the total variance distance between the empirical kernel $\hat{P}$ and the ground-truth $P$ is bounded by $TV(P(\cdot|s, a), \hat{P}(\cdot|s, a)) \leq \delta_g$, then for any policy $\pi : \mathcal{S} \to \mathcal{A}$, we have*

$$\|V_P^\pi - V_{\hat{P}}^\pi\|_\infty \leq \frac{\gamma \delta_g V_{\max}}{(1 - \gamma)},$$

*where $V_{\max} := \frac{r_{\max}}{1 - \gamma}$. Moreover, let $V_P^{\pi^*(P)}$ and $V_{\hat{P}}^{\pi^*(\hat{P})}$ be the corresponding state-values of the optimal policies under the respective kernels. Then we also have*

$$\|V_P^{\pi^*(P)} - V_{\hat{P}}^{\pi^*(\hat{P})}\|_\infty \leq \frac{\gamma \delta_g V_{\max}}{(1 - \gamma)},$$

*where $\pi^*(P)$ and $\pi^*(\hat{P})$ are respectively the optimal policies for the lower-level MDP under $P$ and $\hat{P}$.*

**Proof.** Please see proof details of Lemma 1 in Appendix E. $\square$

We now bound how the configuration errors and estimation errors affect the upper-level rewards. For this analysis, we assume the upper-level MDP adopts a finite number of discrete states and the states

representing $P_c$, $P$ and $\hat{P}$ are the same, namely $s(P_c) = s(P) = s(\hat{P})$, because of their closeness. Here we let $s(P)$ be the upper-level state representation of the lower-level kernel $P$.

**Lemma 2 (Propagated Reward Error Bound)** *We assume $\forall (s, a) \in \mathcal{S} \times \mathcal{A}$, the total variance distance between the empirical kernel $\hat{P}$ and the ideally configured kernel $P_c$ is bounded by $TV(P_c(\cdot|s, a), \hat{P}(\cdot|s, a)) \leq \delta_g + \delta_c$. We assume that for any corresponding pair $(P_c, \hat{P})$, the higher-order transition kernel works the same way, i.e. $Q(s(P_c')|s(P_c), b) = Q(s(\hat{P}')|s(\hat{P}), b), \forall (P_c, \hat{P}), (P_c', \hat{P}') \in \mathcal{P}_c \times \hat{\mathcal{P}}, \forall b \in \mathcal{B}$, and that the cost function works the same way for $P_c$ and $\hat{P}$, i.e. $C(s(P_c), b) = C(s(\hat{P}), b)$. Then for any higher-order deterministic policy $\Theta : \mathcal{P}_c \to \mathcal{B}$, we have the error bound for the upper-level reward function:*

$$\|R_Q^\Theta - \hat{R}_Q^\Theta\|_\infty \leq \frac{\gamma(\delta_g + \delta_c)V_{\max}\|\mu_0\|_\infty}{1 - \gamma}, \tag{8}$$

*where the elements of $\hat{R}_Q^\Theta \in \mathbb{R}^m$ are of the form $R_Q(s(\hat{P}), \Theta(s(\hat{P})))$ (see upper-level reward function definition (3)), the elements of $R_Q^\Theta \in \mathbb{R}^m$ are of the form $R_Q(s(P_c), \Theta(s(P_c)))$, and $m$ is the number of states in the upper-level MDP.*

**Proof.** Please see proof details of Lemma 2 in Appendix F $\qquad\qquad\square$

We now present the error gap for the achievable upper-level state-value function, due to configuration error and estimation errors. We compare the state-value function under ideal configurations and ideal estimation against the state-value function under configuration errors and estimation errors.

For simplicity of presentation, in this lemma and its proof, for the upper-level MDP, we use lower-level transition kernel $P$ (or $P_c$, $\hat{P}$) and its upper-level state representation $s(P)$ (or $s(P_c)$, $s(\hat{P})$) interchangeably.

**Lemma 3 (Error bound Lemma of Bi-level MDPs)** *If $\forall (s, a) \in \mathcal{S} \times \mathcal{A}$, the total variance distance between the empirical kernel $\hat{P}$ and the ideally configured kernel $P_c$ is bounded by $TV(P_c(\cdot|s, a), \hat{P}(\cdot|s, a)) \leq \delta_g + \delta_c$, and if for $\forall (P, b) \in \mathcal{P} \times \mathcal{B}$, the total variance distance between the ground-truth higher-order kernel $Q$ and the empirical higher-order kernel $\hat{Q}$ is bounded by $TV(Q(\cdot|P, b), \hat{Q}(\cdot|P, b)) \leq \Delta$, then for any higher-order policy $\Theta : \mathcal{P} \to \mathcal{B}$, we have that*

$$\|W_Q^\Theta - W_{\hat{Q}}^\Theta\|_\infty \leq \frac{\gamma(\delta_g + \delta_c)V_{\max}\|\mu_0\|_\infty}{(1 - \gamma)(1 - \lambda)} + \frac{2\Delta \cdot \|\mu_0\|_\infty V_{\max} + 2\lambda\Delta \cdot W_{\max}}{1 - \lambda},$$

*where $W_{\max} = \frac{R_{\max}}{1 - \lambda}$.*

**Proof.** Please see proof details of Lemma 3 in Appendix G. $\qquad\qquad\square$

## 6 NUMERICAL EXPERIMENTS

We present two synthetic numerical examples to describe our approaches for solving both the bi-level configurable MDP and the cost-constrained optimization problem formulated on the specific TVCMDP framework. We also conduct experiments in large-scale environments whose dynamics can be explicitly controlled by configuring kernel parameters, including the Cartpole benchmark Brockman et al. (2016) and Block-world Russell & Norvig (2022) environment. In the Cartpole experiment, we employ Deep Q-networks (DQN) to learn lower-level policies corresponding to each uniquely parameterized (configured) environment, while value iteration is used to optimize the upper-level environment parameterization (configuration). Conversely, in the Block-world experiment, we use value iteration to compute the lower-level optimal state-values for each discretized parameter setting, and use DQN in the upper-level to optimize the kernel parameter. These results demonstrate that our proposed framework is adaptable to more complex and continuous RL environments, and that it has potential in realistic applications.

**Continuous configuration on TVCMDP**: We compose a synthetic TVCMDP as described in section 2, *special case*. The composed TVCMDP has 3 states and 2 actions with number of episodes

$K = 2$. In episodes $k = 1$ and $k = 2$, the time-varying kernels $P_1$ and $P_2$ are different. The details of the numerical TVCMDP setting are in Appendix H.1. We solve the cost-constrained optimization problem (4) based on this synthetic example, and optimize the configuration variables $x_1 \in [-1, 1]^{3 \times 3}$ and $x_2 \in [-1, 1]^{3 \times 3}$ under a sequence of budgets, with each budget within the range $[0.5, 14]$. The optimally configured state value averages (blue curve) are shown in Figure 1a. For comparison, we also included the baseline state-value averages (gray dotted line) without any configuration, and the randomly configured state-value averages (orange curve). We observe that the optimally configured approach obviously performs better than the baseline and the randomly configured approach, and the baseline average increases by 23% under configuration. The random configuration variables added to the kernels still satisfy the constraints in (6), but we observe that incorrect configurations may even deteriorate baseline performance, as shown when $B = 3.5$ or $B = 6.5$. This result is insensitive to the parameters such as budget $B$, or configuration constraint parameters $\alpha$ and $\beta$, as shown in Figure 7 in Appendix H.3.

**Bi-level configurable MDP with model-changing actions**: To testify to the feasibility of our bi-level configurable MDP model, we conduct numerical experiments in three different environments. We present the configurable CarPole and Block-world experiments here. Please refer to Appendix H.2 for numerical results of the synthetic bi-level configurable MDP experiment.

*1.Configure the Cartpole baseline*: We construct an upper-level MDP with 4 discrete states by creating four lower-level Cartpole environments, each parametrized (configured) differently. The dynamics are determined by $(g, m_c, m_p, l_p)$: gravity, cart mass, pole mass, and pole length. The four parameter sets $\{(9.8, 1.0, 0.1, 0.5), (9.8, 2.0, 0.1, 0.5), (9.8, 1.0, 0.2, 0.5), (9.8, 1.0, 0.1, 1.0)\}$, correspond to 4 different environments. For each, a DQN agent is trained for 400 episodes to obtain a policy network. The upper-level reward is the performance of a lower-level policy evaluated in a new environment over 20 episodes. The upper-level MDP has four model-changing actions, each deterministically switching to a target environment. Configuration cost depends on which parameter $\{m_c, m_p, l_p\}$ is modified, and is modeled with an exponential; $g$ is excluded due to high cost. Upper-level optimal values are computed via value iteration. Numerical results are shown in 1b.

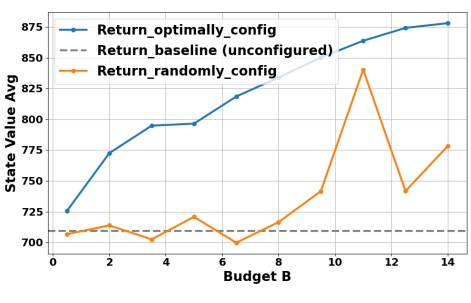

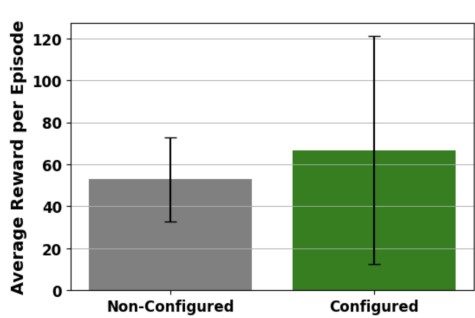

(a) Optimal Continuous Configuration on TVCMDP

(b) Bi-level configuration on the Cartpole environment

Figure 1: Improved returns by continuous and bi-level MDP configurations

*2.Configure the Block-World Environment*: The block-world environment is a grid-based game whose state transition is controlled by the parameter *slip probability* parameter $\alpha$. When the agent chooses an action, there is a probability $\alpha$ that the agent will deviate from the intended direction, potentially moving in an orthogonal direction instead. In the upper-level MDP, the continuous values of $\alpha$ (as a proxy for the lower-level transition kernel) are treated as the upper-level state, and the agent can optimally adjust $\alpha$ using a DQN algorithm. We discretize the continuous parameter $\alpha \in [0, 1]$ with 1000 points, and the corresponding upper-level reward for each lower-level parameter setting is pre-calculated using offline value iteration before training the upper-level DQN. The cost function of configuration action $b$ is $C(\alpha, b) = \mathbb{E}_{Q(\alpha'|\alpha, b)}[\exp(|\alpha' - \alpha|)]$. The training performance of the upper-level DQN, and its evaluation results are presented in Figures 2a and 2b.

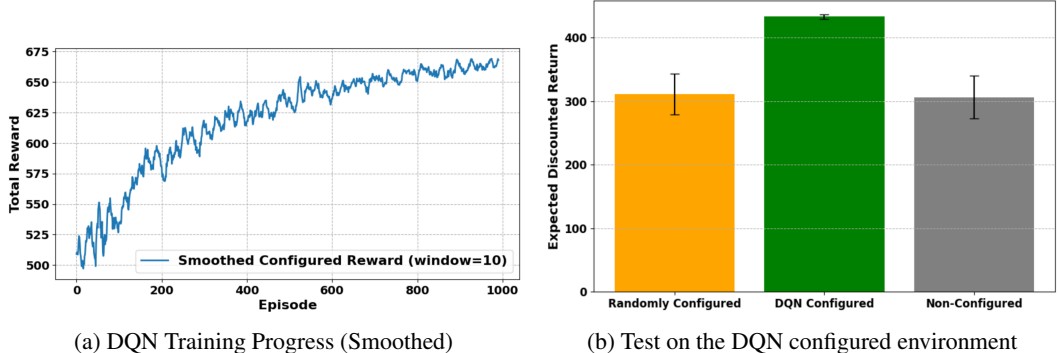

(a) DQN Training Progress (Smoothed)  (b) Test on the DQN configured environment

Figure 2: Bi-level Block-world Configuration: Training and test performances

# 7 CONCLUSION

In this paper, we propose a new framework in the context of configurable reinforcement learning
in which an agent can actively change the environment while simultaneously optimizing its policy
in a time-varying setting. We formalize this idea through the multi-layer configurable time-varying
Markov decision process (TVCMDP). Within this framework, we analyze two important cases:
the bi-level configurable MDP and the time-variant MDP with continuous configuration. For each
setting, we develop solution algorithms by approaches like convex optimization and bi-level value-
iteration, and provide an error analysis on the performance of these algorithms. In the future, a
natural next step is to extend our model-based algorithms to fully model-free algorithms for con-
figurable time-varying MDPs and evaluate their effectiveness in larger-scale environments. Another
promising direction is the study of reward design with configuration cost, which would allow a
principled trade-off between reward gains and the cost of modifying the environment.

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

## A    MOTIVATING EXAMPLES

**Transportation infrastructure**: Consider another example of transportation infrastructure. In this example, model-changing actions are the actions that build, modify, or improve the transportation infrastructures in a city. The states are traffic congestion and transportation safety situations in the city. These model-changing actions will dictate the transition kernels between different traffic congestion and transportation safety situations. The upper-level MDP is for learning how to change the transportation infrastructure to lead to better transition kernels for the lower-level MDP for traffic. Note that over time, transportation infrastructures can deteriorate, and this is one example of time-varying MDP configurations, which will be discussed later.

**Training of Drones**: In another example, let us consider a collection of unmanned drones performing a certain task. The task needs the unmanned drones to perform sensing, communication, and control cooperatively. Without manned aircrafts, which are more capable of sensing, payload carrying, and communication, the unmanned drones may be in an unfavorable environment. We can configure a favorable environment for the unmanned drones by using manned aircrafts to transport the unmanned drones to the desirable location for the task, and to configure sensing and communication infrastructure for the unmanned drones. The configuration actions using manned aircraft can lead to random transitions between configurations, depending on random external factors such as external interference and the fates of the manned aircrafts sent to configure the environment. The MDP for manned aircrafts to configure the environment will work as an upper-level MDP, and the adaptations of unmanned drones are governed by the lower-level MDP.

**Trading market**: Another example is when the central bank adjusts the short-term interest rate (i.e., the federal funds rate), this upper-level action directly influences the market dynamics and ultimately affects the stock valuation. A widely used framework for stock valuation is the discounted cash flow model (DCF), which estimates a firm's value based on its expected future cash flows discounted by the weighted average cost of capital (WACC). Since the interest rate is a key component of WACC, an increase in the interest rate raises the discount factor, thereby reducing the firm's present stock valuation.

## B    RELATED WORKS

Our work is different from hierarchical RL/MDP Li et al. (2022) because in our multilayer model, the upper-level MDP is built upon choosing (configuring) a better transition kernel, while hierarchical MDP deals with decomposing long-horizon tasks into simpler subtasks or learning hierarchical policies. For other literature on non-stationary environments, such as semi-MDP Sutton et al. (1999), Meta-RL Duan et al. (2016) and Bayesian MDPs Duff & Barto (2002), the key distinction from our formulation lies in the design-driven "configuration" nature of configurable MDPs: the agent is allowed to modify the environment itself to improve its potential returns. In semi-MDPs, the agent can only temporally extend actions, but the environment remains fixed. In contrast, in a configurable MDP the agent can choose both the environment configuration and an associated policy. Meta-RL aims to train the agent across a distribution of tasks to enable rapid adaptation to unseen tasks, while the configurable MDP assumes a known set of configurations, and the agent's goal is to pick the best environment and policy.

## C    JACOBIAN OF STATE-VALUE FUNCTIONS

We denote $V_i^\pi = V^\pi$ as the $i$-th element of the state-values, $P_i^\pi$ as the $i$-th row of $P^\pi$, and $P_{ij}^\pi$ as the $(i, j)$-th element of $P^\pi$.

We assume that for a sufficiently small configuration $x \in [-1, 1]^{n \times n}$ on the transition kernel $P^\pi$, the policy applied to the configured kernel $P^\pi + x$ remains unchanged as $\pi$. We fix the discount factor $\gamma$ (scalar) and the reward vector $r^\pi$, and we let the optimal transition kernel $P^\pi$ change by x. Then, $V^\pi$ is a function of $P^\pi$. The Jacobian of $V^\pi$ with respect to $P^\pi$ is denoted as $\nabla_{P^\pi} V^\pi$, which

is a tensor in $\mathbb{R}^{n \times n \times n}$. We write out the Jacobian in the following form:

$$
\nabla_{P^\pi} V^\pi = \begin{bmatrix} \frac{\partial V_1^\pi}{\partial P^\pi} \\ \vdots \\ \frac{\partial V_n^\pi}{\partial P^\pi} \end{bmatrix} = \begin{bmatrix} \begin{bmatrix} \frac{\partial V_1^\pi}{\partial P_{11}^\pi}, \dots, \frac{\partial V_1^\pi}{\partial P_{1n}^\pi} \\ \vdots \\ \frac{\partial V_1^\pi}{\partial P_{n1}^\pi}, \dots, \frac{\partial V_1^\pi}{\partial P_{nn}^\pi} \end{bmatrix} \\ \vdots \\ \begin{bmatrix} \frac{\partial V_n^\pi}{\partial P_{11}^\pi}, \dots, \frac{\partial V_n^\pi}{\partial P_{1n}^\pi} \\ \vdots \\ \frac{\partial V_n^\pi}{\partial P_{n1}^\pi}, \dots, \frac{\partial V_n^\pi}{\partial P_{nn}^\pi} \end{bmatrix} \end{bmatrix} . \tag{9}
$$

To solve for the Jacobian $\nabla_{P^\pi} V^\pi$, we let x be small enough and write the right-hand side of (7), and here $\mathrm{x}_{*i} \in [-1,1]^n$ is the $i$-th column of x, $N_i$ is a scalar

$$
N + M \begin{bmatrix} \mathrm{x}_{11}, \dots, \mathrm{x}_{1n} \\ \mathrm{x}_{21}, \dots, \mathrm{x}_{2n} \\ \vdots \\ \mathrm{x}_{n1}, \dots, P_{nn} \end{bmatrix} \begin{bmatrix} N_1 \\ N_2 \\ \vdots \\ N_n \end{bmatrix}
$$

$$
= N + M \left[ N_1 \begin{bmatrix} | \\ \mathrm{x}_{*1} \\ | \end{bmatrix} + N_2 \begin{bmatrix} | \\ \mathrm{x}_{*2} \\ | \end{bmatrix} + \dots + N_n \begin{bmatrix} | \\ \mathrm{x}_{*n} \\ | \end{bmatrix} \right]
$$

$$
= N + \left[ N_1 M \begin{bmatrix} | \\ \mathrm{x}_{*1} \\ | \end{bmatrix} + N_2 M \begin{bmatrix} | \\ \mathrm{x}_{*2} \\ | \end{bmatrix} + \dots + N_n M \begin{bmatrix} | \\ \mathrm{x}_{*n} \\ | \end{bmatrix} \right]
$$

We show the process of solving for the gradient of $V_i^\pi$, denoted as $\frac{\partial V_i^\pi}{\partial P^\pi}$, as an example. Here $\frac{\partial V_i^\pi}{\partial P^\pi}$ is the $i$-th element of the Jacobian $\nabla_{P^\pi} V^\pi$ defined in (9). Based on previous derivations, we have

$$
V_i^\pi(P^\pi + \mathrm{x}) - V_i^\pi(P^\pi) \approx \sum_{j=1}^n N_j M_{i*} \mathrm{x}_{*j}.
$$

Here $N_j$ is a scalar, $M_{i*} \in \mathbb{R}^n$ is the $i$-th row vector of $M$, and $\mathrm{x}_{*j} \in [-1,1]^n$ is the $j$-th column vector of x. The gradient $\frac{\partial V_i^\pi}{\partial P^\pi}$ is therefore

$$
\frac{\partial V_i^\pi}{\partial P^\pi} = \begin{bmatrix} \frac{\partial V_i^\pi}{\partial P_{11}^\pi}, \dots, \frac{\partial V_i^\pi}{\partial P_{1n}^\pi} \\ \vdots \\ \frac{\partial V_i^\pi}{\partial P_{n1}^\pi}, \dots, \frac{\partial V_i^\pi}{\partial P_{nn}^\pi} \end{bmatrix} = \begin{bmatrix} N_1 M_{i1} & N_2 M_{i1} & \dots & N_n M_{i1} \\ N_1 M_{i2} & N_2 M_{i2} & \dots & N_n M_{i2} \\ \vdots \\ N_1 M_{in} & N_2 M_{in} & \dots & N_n M_{in} \end{bmatrix} = \begin{bmatrix} | & | & & | \\ M_{i*} & M_{i*} & (n \text{ cols.}) & M_{i*} \\ | & | & & | \end{bmatrix} N
$$

Let $E_i \in \mathbb{R}^{n \times n}$ denote the matrix that has $M_{i*}$ as its repeated $n$ columns. Therefore, the Jacobian of $V^\pi$ can be represented as:

$$
\nabla_{P^\pi} V^\pi = [E_1 N, E_2 N, \dots, E_n N]^T \tag{10}
$$

## D  ALGORITHM: MODEL-BASED BI-LEVEL VALUE ITERATION

To estimate each ground-truth lower-level kernel $P \in \mathcal{P}$, given a dataset $D$ of trajectories, $D = \{(s_1, a_1, r_1, s_2, \dots, s_{T+1})\}$, and we convert it into a series of $\{(s, a, r, s')\}$ tuples. We break each trajectory into $T$ tuples: $(s_1, a_1, r_1, s_2), (s_2, a_2, r_2, s_3), \dots, (s_T, a_T, r_T, s_{T+1})$. For every state-action pair $(s, a)$, let $D_{(s,a)}$ be the subset of tuples where the first element of the tuple is $s$, and the second element of the tuple is $a$. Then the elements in $D_{(s,a)}$ can be represented by $(r, s')$ since they share the same state-action pair $(s, a)$. Each empirical lower-level kernel $\hat{P}$ is estimated by the empirical frequency of state transitions, i.e., $\forall (s, a) \in \mathcal{S} \times \mathcal{A}, \ \hat{P}(s'|s, a) = Count((r, s'))/\left| D_{(s,a)} \right|$.

Similarly, to estimate the ground-truth upper-level kernel $Q$, given a dataset $D^q$ of trajectories, $D^q = \{(P_1, b_1, R_1, P_2, \ldots, P_{K+1})\}$, and we convert it into a series of $\{(P, b, R, P')\}$ tuples. We break each trajectory into $K$ tuples: $(P_1, b_1, R_1, P_2), (P_2, b_2, R_2, P_3), \ldots, (P_K, b_K, R_K, P_{K+1})$. For every lower-level kernel (upper-level state) and model-changing-action pair $(P, b)$, let $D^q_{(P,b)}$ be the subset of tuples where the first element of the tuple is $P$, and the second element of the tuple is $b$. Then the elements in $D^q_{(P,b)}$ can be represented by $(R, P')$. The empirical upper-level kernel $\hat{Q}$ is estimated by the empirical frequency of kernel transitions, i.e., $\forall (P, b) \in \mathcal{P} \times \mathcal{B}, \ \hat{Q}(P'|P, b) = Count((R, P'))/\left|D^q_{(P,b)}\right|$.

With the estimations $\{\hat{P}\}_{\hat{P} \in \hat{\mathcal{P}}}$ and $\hat{Q}$, we provide the model-based bi-level value iteration algorithm as follows:

---

**Algorithm 1** Bi-level Value Iteration

---

**Input**: The $m$ empirical infinite horizon lower-level MDPs $\hat{\mathcal{M}}_L = \{\mathcal{S}, \mathcal{A}, \hat{P}, \mu_0, r, \gamma\}$, for $\hat{P} \in \hat{\mathcal{P}}$, and the infinite-horizon empirical upper-level MDP $\hat{\mathcal{M}}_U = \{\hat{\mathcal{P}}, \mathcal{B}, \hat{Q}, \hat{R}, \lambda\}$, number of lower-level iterations $H_L > 0$, number of upper-level iterations $H_U > 0$, initial estimations $\{V^0_{\hat{P}}\}_{\hat{P} \in \hat{\mathcal{P}}}$ and $W^0$

**Result**: upper-level estimation $W^H \in \mathbb{R}^m$ and higher-order policy $\Theta_H$

1: **for** $h$ from 0 to $H_L$ **do**                                    ▷ Lower-level value iterations
2:     **for** $\hat{P} \in \hat{\mathcal{P}}$, **do**
3:         $V^{h+1}_{\hat{P}}(s) \leftarrow \max_{a \in \mathcal{A}} \left( r(s, a) + \gamma \sum_{s' \in \mathcal{S}} \hat{P}(s'|s, a) V^h_{\hat{P}}(s') \right), \forall s \in \mathcal{S}$
4:     **end for**
5:     $\pi^{H_L}_{\hat{P}} \leftarrow$ greedy policy with respect to $V^{H_L}_{\hat{P}}$
6:     $J_{\hat{P}} \leftarrow \mu_0^T V^{H_L}_{\hat{P}}$
7: **end for**
8: **for** $h$ from 0 to $H_U$ **do**                                  ▷ upper-level value iteration
9:     $W^{h+1}(\hat{P}) \leftarrow \max_{b \in \mathcal{B}} \left( \sum_{\hat{P}' \in \hat{\mathcal{P}}} \hat{Q}(\hat{P}'|\hat{P}, b) \left( J_{\hat{P}'} + \lambda W^h(\hat{P}') \right) \right), \forall \hat{P} \in \hat{\mathcal{P}}$
10: **end for**
11: $\Theta^{H_U} \leftarrow$ greedy policy with respect to $W^{H_U}$
12: return $\Theta_{H_U}, W^{H_U}$

---

# E    PROOF OF LEMMA 1

**Proof.** The first part of this proof mostly follows similar steps in Sun (2021). For any state $s \in \mathcal{S}$ and for any policy $\pi$, let $r^\pi(s)$ denote $r(s, \pi(s))$, and $P^\pi(s)$ denote $P(\cdot|s, \pi(s))$ (as a row vector):

$$
\begin{aligned}
\left| V^\pi_P(s) - V^\pi_{\hat{P}}(s) \right| &= \left| r^\pi(s) + \gamma P^\pi(s) V^\pi_P - \left( r^\pi(s) + \gamma \hat{P}^\pi(s) V^\pi_{\hat{P}} \right) \right| \\
&= \gamma \left| P^\pi(s) V^\pi_P - \hat{P}^\pi(s) V^\pi_{\hat{P}} \right| \\
&= \gamma \left| P^\pi(s) V^\pi_P - P^\pi(s) V^\pi_{\hat{P}} + P^\pi(s) V^\pi_{\hat{P}} - \hat{P}^\pi(s) V^\pi_{\hat{P}} \right| \\
&\leq \gamma \left| P^\pi(s) \left( V^\pi_P - V^\pi_{\hat{P}} \right) \right| + \gamma \left| \left( P^\pi(s) - \hat{P}^\pi(s) \right) V^\pi_{\hat{P}} \right| \\
&\leq \gamma \left\| V^\pi_P - V^\pi_{\hat{P}} \right\|_\infty + \gamma \left| \left( P^\pi(s) - \hat{P}^\pi(s) \right) V^\pi_{\hat{P}} \right| \\
&= \gamma \left\| V^\pi_P - V^\pi_{\hat{P}} \right\|_\infty + \gamma \left| \left( P^\pi(s) - \hat{P}^\pi(s) \right) \left( V^\pi_{\hat{P}} - \frac{V_{\max}}{2} \cdot \mathbf{1} \right) \right| \\
&\leq \gamma \left\| V^\pi_P - V^\pi_{\hat{P}} \right\|_\infty + \gamma \left\| \left( P^\pi(s) - \hat{P}^\pi(s) \right) \right\|_1 \left\| V^\pi_{\hat{P}} - \frac{V_{\max}}{2} \right\|_\infty \\
&\leq \gamma \left\| V^\pi_P - V^\pi_{\hat{P}} \right\|_\infty + \gamma \delta_g V_{\max}.
\end{aligned}
$$

---

Since the above inequality holds for all $s \in \mathcal{S}$, we have that

$$\left\| V_P^\pi - V_{\hat{P}}^\pi \right\|_\infty \leq \gamma \left\| V_P^\pi - V_{\hat{P}}^\pi \right\|_\infty + \gamma \delta_g V_{\max},$$

$$\left\| V_P^\pi - V_{\hat{P}}^\pi \right\|_\infty \leq \frac{\gamma \delta_g V_{\max}}{1 - \gamma}.$$

Let us now prove the second claim. For any state $s$ in the lower-level MDP, by taking $\pi = \pi^*(P)$, we have

$$V_P^{\pi^*(P)}(s) \leq V_{\hat{P}}^{\pi^*(P)}(s) + \frac{\gamma \delta_g V_{\max}}{(1 - \gamma)} \leq V_{\hat{P}}^{\pi^*(\hat{P})}(s) + \frac{\gamma \delta_g V_{\max}}{(1 - \gamma)},$$

where the 2nd inequality is due to $V_{\hat{P}}^{\pi^*(P)}(s) \leq V_{\hat{P}}^{\pi^*(\hat{P})}(s)$. By symmetry, we also have

$$V_{\hat{P}}^{\pi^*(\hat{P})}(s) \leq V_P^{\pi^*(\hat{P})}(s) + \frac{\gamma \delta_g V_{\max}}{(1 - \gamma)} \leq V_P^{\pi^*(P)}(s) + \frac{\gamma \delta_g V_{\max}}{(1 - \gamma)},$$

thus the 2nd claim follows.

$\square$

## F   PROOF OF LEMMA 2

**Proof.** For a lower-level transition kernel $P$, we define $J(\pi^*(P), P)$ as the "reward" $J$ reported by the lower-level MDP to the upper-level MDP (as $J$ in Algorithm 1) if the lower-level MDP has transition kernel $P$ and adopts the optimal policy $\pi^*(P)$. For any higher-order policy $\Theta$, and for any $P_c$ and its corresponding estimate $\hat{P}$, with the definition of the upper-level reward function (3), we have that

$$\left| R_Q(s(P_c), \Theta(s(P_c))) - R_Q(s(\hat{P}), \Theta(s(\hat{P}))) \right|$$

$$= \left| \sum_{s(P_c')} Q(s(P_c')|s(P_c), \Theta(s(P_c))) J(\pi^*(P_c'), P_c') - \right.$$

$$\left. \sum_{s(\hat{P}')} Q(s(\hat{P}')|s(\hat{P}), \Theta(s(\hat{P}))) J(\pi^*(\hat{P}'), \hat{P}') \right|$$

$$= \left| \sum_{s(P_c')=s(\hat{P}')} Q(s(P_c')|s(P_c), \Theta(s(P_c))) \left( J(\pi^*(P_c'), P_c') - J(\pi^*(\hat{P}'), \hat{P}') \right) \right|$$

$$\leq \left| \sum_{s(P_c')} Q(s(P_c')|s(P_c), \Theta(s(P_c))) \right| \max_{P_c', \hat{P}', s(P_c')=s(\hat{P}')} \left| J(\pi^*(P_c'), P_c') - J(\pi^*(\hat{P}'), \hat{P}') \right|$$

$$= \max_{P_c', \hat{P}', s(P_c')=s(\hat{P}')} \left| J(\pi^*(P_c'), P_c') - J(\pi^*(\hat{P}'), \hat{P}') \right|$$

$$= \max_{P_c', \hat{P}', s(P_c')=s(\hat{P}')} \left| \mu_0^T \left( V_{P_c'}^{\pi^*(P_c')} - V_{\hat{P}'}^{\pi^*(\hat{P}')} \right) \right|$$

$$\leq \max_{P_c', \hat{P}', s(P_c')=s(\hat{P}')} \| \mu_0 \|_\infty \left\| \left( V_{P_c'}^{\pi^*(P_c')} - V_{\hat{P}'}^{\pi^*(\hat{P}')} \right) \right\|_\infty$$

$$\leq \frac{\gamma(\delta_g + \delta_c) V_{\max} \| \mu_0 \|_\infty}{1 - \gamma}.$$

Note that in the derivations above, we have $s(P_c') = s(\hat{P}')$ due to the assumption that these two lower-level kernels have the same state representation in the upper-level MDP. The last inequality can be achieved by directly applying Lemma 1. Since the above inequality holds for all the ideal-estimated pair $(P_c, \hat{P}) \in \mathcal{P}_c \times \hat{\mathcal{P}}$, we have that

$$\| R_Q^\Theta - \hat{R}_Q^\Theta \|_\infty \leq \frac{\gamma(\delta_g + \delta_c) V_{\max} \| \mu_0 \|_\infty}{1 - \gamma}.$$

$\square$

# G    PROOF OF LEMMA 3

**Proof.** For any ideal $P_c$, and its estimate $\hat{P}$, using Bellman equation $W_Q^\Theta = R_Q(P_c, \Theta(P_c)) + \lambda Q^\Theta(P_c)W_Q^\Theta$, we have

$$\left| W_Q^\Theta(P_c) - W_{\hat{Q}}^\Theta(\hat{P}) \right| \leq \underbrace{\left| R_Q(P_c, \Theta(P_c)) - R_{\hat{Q}}(\hat{P}, \Theta(\hat{P})) \right|}_{①} + \lambda \underbrace{\left| Q^\Theta(P_c)W_Q^\Theta - \hat{Q}^\Theta(\hat{P})W_{\hat{Q}}^\Theta \right|}_{②},$$

(11)

where $Q^\Theta(P_c) = Q(\cdot|P_c, \Theta(P_c))$, and $\hat{Q}^\Theta(\hat{P}) = \hat{Q}(\cdot|\hat{P}, \Theta(\hat{P}))$.

To bound ①, we have the following derivations. Notice that $Q^\Theta(\cdot) \in \mathbb{R}^m$ and $\hat{Q}^\Theta(\cdot) \in \mathbb{R}^m$ are both row vectors, and $Q^\Theta(P_c) = Q(\cdot|P_c, \Theta(P_c))$ means the ground-truth distribution of the next upper-level state given the current state is $P_c$ and the agent follows the higher-order policy $\Theta$. Similarly, $\hat{Q}^\Theta(\hat{P}) = \hat{Q}(\cdot|\hat{P}, \Theta(\hat{P}))$ is the empirical distribution of the next upper-level state given the current state $\hat{P}$ and the agent follows the higher-order policy $\Theta$.

$$\left| R_Q(P_c, \Theta(P_c)) - R_{\hat{Q}}(\hat{P}, \Theta(\hat{P})) \right|$$

$$\leq \left| R_Q(P_c, \Theta(P_c)) - R_Q(\hat{P}, \Theta(\hat{P})) \right| + \left| R_Q(\hat{P}, \Theta(\hat{P})) - R_{\hat{Q}}(\hat{P}, \Theta(\hat{P})) \right|$$

$$\leq \frac{\gamma(\delta_g + \delta_c)V_{\max}\|\mu_0\|_\infty}{1 - \gamma} + \left| R_Q(\hat{P}, \Theta(\hat{P})) - R_{\hat{Q}}(\hat{P}, \Theta(\hat{P})) \right| \text{ (note: by Lemma 2)}$$

$$= \frac{\gamma(\delta_g + \delta_c)V_{\max}\|\mu_0\|_\infty}{1 - \gamma} + \left| \sum_{\hat{P}'} (Q - \hat{Q})(\hat{P}'|\hat{P}, \Theta(\hat{P}))J(\pi^*(\hat{P}'), \hat{P}') \right|$$

$$\leq \frac{\gamma(\delta_g + \delta_c)V_{\max}\|\mu_0\|_\infty}{1 - \gamma} + \left| \sum_{\hat{P}'} (Q - \hat{Q})(\hat{P}'|\hat{P}, \Theta(\hat{P})) \right| \max_{\hat{P}'} \left| J(\pi^*(\hat{P}'), \hat{P}') \right|$$

$$= \frac{\gamma(\delta_g + \delta_c)V_{\max}\|\mu_0\|_\infty}{1 - \gamma} + 2\Delta\|\mu_0\|_\infty V_{\max}.$$

To bound ②, we have that (we disregard $\lambda$ for now)

$$\left| Q^\Theta(P_c)W_Q^\Theta - \hat{Q}^\Theta(\hat{P})W_{\hat{Q}}^\Theta \right| \leq \underbrace{\left| Q^\Theta(P_c)W_Q^\Theta - Q^\Theta(\hat{P})W_Q^\Theta \right|}_{0} + \left| Q^\Theta(\hat{P})W_Q^\Theta - \hat{Q}^\Theta(\hat{P})W_{\hat{Q}}^\Theta \right|$$

$$= \left| Q^\Theta(\hat{P})W_Q^\Theta - \hat{Q}^\Theta(\hat{P})W_{\hat{Q}}^\Theta \right|$$

$$\leq \left| Q^\Theta(\hat{P})W_Q^\Theta - Q^\Theta(\hat{P})W_{\hat{Q}}^\Theta \right| + \left| Q^\Theta(\hat{P})W_{\hat{Q}}^\Theta - \hat{Q}^\Theta(\hat{P})W_{\hat{Q}}^\Theta \right|$$

$$\leq \left| Q^\Theta(\hat{P})\left(W_Q^\Theta(P_c) - W_{\hat{Q}}^\Theta(\hat{P})\right) \right| + \left\| (Q^\Theta - \hat{Q}^\Theta)(\hat{P}) \right\|_1 \left\| W_{\hat{Q}}^\Theta \right\|_\infty$$

$$= \left| Q^\Theta(\hat{P})\left(W_Q^\Theta(P_c) - W_{\hat{Q}}^\Theta(\hat{P})\right) \right| + 2\Delta W_{\max}$$

$$\leq \left\| Q^\Theta(\hat{P}) \right\|_1 \left\| W_Q^\Theta(P_c) - W_{\hat{Q}}^\Theta(\hat{P}) \right\|_\infty + 2\Delta W_{\max}$$

$$= \left\| W_Q^\Theta(P_c) - W_{\hat{Q}}^\Theta(\hat{P}) \right\|_\infty + 2\Delta W_{\max},$$

where the first term on the righthand side of the first inequality is 0 because with $s(P_c) = s(\hat{P})$, $Q^\Theta(P_c) = Q^\Theta(\hat{P})$.

By reorganizing terms, our goal (11) is bounded by:

$$\left| W_Q^\Theta(P_c) - W_{\hat{Q}}^\Theta(\hat{P}) \right| \leq \underbrace{\frac{\gamma(\delta_g + \delta_c)V_{\max}\|\mu_0\|_\infty}{1 - \gamma} + 2\Delta\|\mu_0\|_\infty V_{\max}}_{①}$$

$$+ \lambda \underbrace{\left\| W_Q^\Theta(P_c) - W_{\hat{Q}}^\Theta(\hat{P}) \right\|_\infty + 2\lambda\Delta W_{\max}}_{②}.$$

Since the above inequality holds for all $(P_c, \hat{P}) \in \mathcal{P}_c \times \hat{\mathcal{P}}$, we have

$$\left\| W_Q^\Theta - W_{\hat{Q}}^\Theta \right\|_\infty \leq \frac{\gamma(\delta_g + \delta_c)V_{\max}\|\mu_0\|_\infty}{(1-\gamma)(1-\lambda)} + \frac{2\Delta \cdot \|\mu_0\|_\infty V_{\max} + 2\lambda\Delta \cdot W_{\max}}{1-\lambda}.$$

$\square$

## H  NUMERIC SETTINGS

### H.1  SYNTHETIC TVCMP

We are considering a time-varying configurable MDP with the state space $\mathcal{S} = \{0, 1, 2\}$, action space $\mathcal{A} = \{\text{left, right, stay}\}$, and the number of episodes is $K = 2$. The time varying transition kernel $P_1$ and $P_3$ in episodes $k = 1$ and $k = 2$ are, respectively,

$$P_1^{(l)} = \begin{bmatrix} 0 & 0.15 & 0.85 \\ 0.75 & 0 & 0.25 \\ 0.25 & 0.75 & 0 \end{bmatrix}, \quad P_1^{(r)} = \begin{bmatrix} 0 & 0.85 & 0.15 \\ 0.15 & 0 & 0.85 \\ 0.85 & 0.15 & 0 \end{bmatrix}, \quad P_1^{(s)} = \begin{bmatrix} 0.9 & 0.05 & 0.05 \\ 0.05 & 0.9 & 0.05 \\ 0.05 & 0.05 & 0.9 \end{bmatrix}$$

$$P_2^{(l)} = \begin{bmatrix} 0 & 0.45 & 0.55 \\ 0.65 & 0 & 0.35 \\ 0.45 & 0.55 & 0 \end{bmatrix}, \quad P_2^{(r)} = \begin{bmatrix} 0 & 0.75 & 0.25 \\ 0.25 & 0 & 0.75 \\ 0.85 & 0.15 & 0 \end{bmatrix}, \quad P_2^{(s)} = \begin{bmatrix} 0.8 & 0.1 & 0.1 \\ 0.2 & 0.6 & 0.2 \\ 0.05 & 0.05 & 0.9 \end{bmatrix}.$$

For each action $a$, the transition probabilities are given by the matrix $P_i^{(a)} \in \mathbb{R}^{3\times 3}, i = 1, 2$, where the rows index the current state $s$ and the columns index the next state $s'$. The initial state distribution of every episode $\mu_0 = [1/3, 1/3, 1/3]^T$ is uniform. The reward function $r$ remains the same in all episodes. $r(s, a)$ is defined for each state and action pair and is represented as the following matrix:

$$r(s, a) = \begin{bmatrix} 10 & 5 & 1 \\ 2 & 20 & 10 \\ 20 & 4 & 40 \end{bmatrix}.$$

Here, rows correspond to states, and columns correspond to actions. $\gamma = 0.9$. The configuration budgets considered include $[0.5, 2.06, 3.61, 5.17, 6.72, 8.28, 9.83, 11.39, 12.94, 14.0]$.

### H.2  SYNTHETIC BI-LEVEL MDP

**Synthetic setting**: We give the numeric settings of the synthetic bi-level MDP. In the lower-level MDP, the state space $\mathcal{S}$ consists of two dimensions: (price-level, portfolio). The price-level has three statuses: $\{0:\text{Low}, 1:\text{Neutral}, 2:\text{High}\}$. Each price level corresponds to a price in $(90, 100, 130)$. The portfolio has two statuses: $\{0:\text{Cash}, 1:\text{Holding}\}$, so there are total 6 states in $\mathcal{S}$. The lower-level action space $\mathcal{A} = \{\text{buy, sell}\}$. There are 3 modes of lower-level transition kernels, or equivalently, 3 states in the upper-level MDP, $P_1, P_2, P_3$, which respectively determine the price-level transitions during "boom", "recession", and "stabilization". We set them as

$$P_1 = \begin{bmatrix} 0.6 & 0.3 & 0.1 \\ 0.4 & 0.4 & 0.2 \\ 0.3 & 0.5 & 0.2 \end{bmatrix}, \quad P_2 = \begin{bmatrix} 0.2 & 0.5 & 0.3 \\ 0.1 & 0.6 & 0.3 \\ 0.05 & 0.25 & 0.7 \end{bmatrix} \quad P_3 = \begin{bmatrix} 0.2 & 0.6 & 0.2 \\ 0.2 & 0.6 & 0.2 \\ 0.1 & 0.5 & 0.4 \end{bmatrix}.$$

The transitions between the status of the portfolio depend on the action. If $s = (\cdot, 0)$ and $a = \text{buy}$, then $s' = (\cdot, 1)$; If $s = (\cdot, 1)$ and $a = \text{sell}$, then $s' = (\cdot, 0)$. The reward function $r$ is

$$r(s, a) = \begin{cases} -1, \text{if } s = (\cdot, 0), a = \text{ buy} \\ \text{new price - old price} - 1, \text{if } s = (\cdot, 1), a = \text{ sell} \\ \text{new price - old price, if } s = (\cdot, 1) \end{cases}$$

$\gamma = 0.95$. $\mu_0$ is uniformly $1/6$. In the upper-level, the state space $\mathcal{P} = \{P_1, P_2, P_3\}$, the action space is $\mathcal{B} = \{0:\text{Decrease rate, 1:Increase rate, 2: Keep rate}\}$. The upper-level kernel governs the transitions between lower kernels $P_i$ and is given by:

$$Q^{(De)} = \begin{bmatrix} 0.7 & 0.2 & 0.1 \\ 0.6 & 0.2 & 0.2 \\ 0.7 & 0.1 & 0.2 \end{bmatrix} \quad Q^{(In)} = \begin{bmatrix} 0.5 & 0.3 & 0.2 \\ 0.3 & 0.5 & 0.2 \\ 0.4 & 0.4 & 0.2 \end{bmatrix} \quad Q^{(Ke)} = \begin{bmatrix} 0.6 & 0.25 & 0.15 \\ 0.4 & 0.4 & 0.2 \\ 0.2 & 0.3 & 0.5 \end{bmatrix}$$

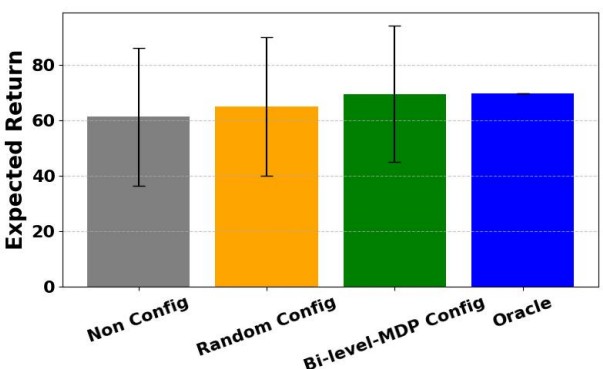

Figure 3: Bi-level configuration on the synthetic example

$\lambda = 0.95$. The upper-level reward $R$ is computed according to 3. The configuration cost function is determined by the current upper-level state and configuration action, and we set it to be:

$$C(P, b) = \begin{bmatrix} 0.2 & 0.1 & 0.05 \\ 0.5 & 0.3 & 0.1 \\ 0.3 & 0.2 & 0.1 \end{bmatrix},$$

with the rows index kernel mode, and the columns index upper-level actions.

**Synthetic Bi-level configurable MDP Experiment**: We compose a synthetic bi-level configurable MDP as described in 2, *Bi-level Configurable MDP*. The upper-level MDP has 3 discrete states (equivalently, there are 3 discrete lower-level transition kernels) and 3 model-changing actions, while each lower-level MDP has 3 states and 2 actions. The details of the environment setting are given in the Appendix H.2. In Figure 3, we show that throughout 100 test episodes, the agent achieves a higher average return (green bar) when following the optimal upper-level policy derived from Algorithm 1, compared to the two alternative approaches. For comparison, we also computed the average return under the non-configuration mode (gray bar), where the lower-level transition kernel is uniformly sampled over the 100 test episodes and the agent just follows the corresponding primitive optimal policy within each episode, the random configuration mode (orange bar), where model-changing actions are chosen randomly, and the oracle (blue bar), where the lower-level kernel is fixed to be the optimal one and the agent uses the corresponding optimal policy over the 100 test episodes. Our bi-level MDP configuration obviously shows better performance than the non-configuration mode and the random configuration mode, and its performance is the closest to the oracle, as expected.

**Comparison between the theoretical and empirical error bounds**: In this synthetic Bi-level MDP, we separately perturb: 1) the lower-level transition kernel $P(\cdot|s, a), \forall s, a$ using $\delta_g + \delta_c \in [0, 0.25]$ and we denote the noisy lower-level kernel with $P'$, and 2) the upper-level model kernel $Q(\cdot|P, b), \forall P, b$ using $\Delta \in [0, 0.025]$, and we denote the noisy upper-level kernel with $Q'$. This allows us to isolate how lower-level and upper-level errors influence the performance of Algorithm 1.

Figure 4 reports the resulting lower-level state value errors between $P$ and $P'$. The blue curve shows the infinity norm difference between the true and perturbed lower-level value functions. We sample noisy lower-level kernels $P'$ from the distribution of noisy kernels such that $TV(P, P') = \delta_g + \delta_c$ and record the maximum difference between state values in terms of $\max_{P'} \|V^{P,\pi^*} - V^{P',\pi^*}\|_\infty$ among all the realizations of the perturbations. The error grows linearly with the perturbation size and stays within the theoretical bound.

Figure 5 reports the resulting upper-level state value errors separately by perturbing the lower-level kernel and the upper-level kernel. In the first case, $\Delta = 0$, and in the second case, $\delta_g + \delta_c = 0$. The blue curves show the infinity norm difference between the true and perturbed upper-level value functions. The error grows linearly with the two perturbation sizes separately and stays within the theoretical bound stated in Lemma 3.

**Comparison with classic configurable MDPs**: The key distinction between our bi-level configurable MDP and the classical CMDP formulations, such as gradient CMDP and Stackelberg CDMP,

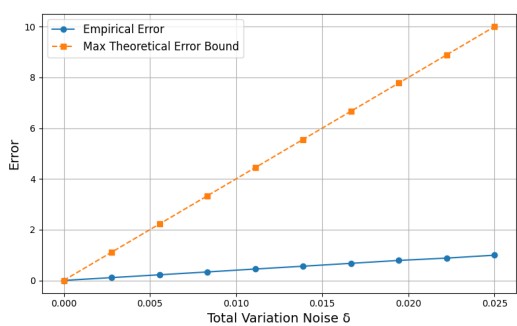

Figure 4: Comparison of error bounds, lower-level state value

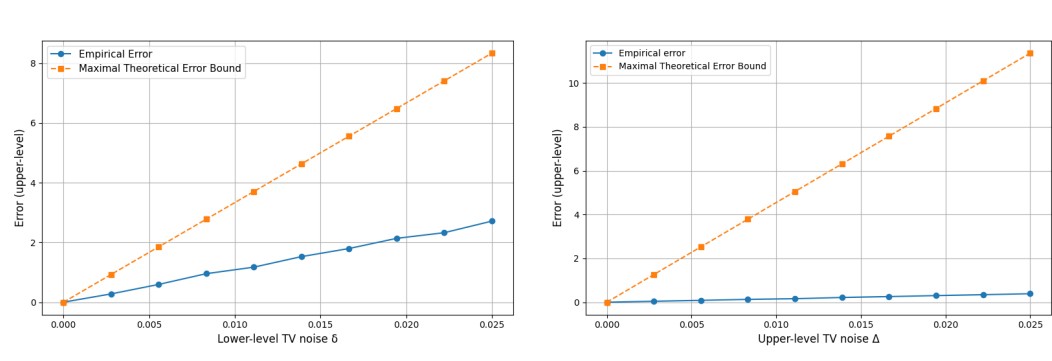

(a) Total variation noise on the lower-level kernel

(b) Total variation noise on the upper-level kernel

Figure 5: Comparison between theoretical and empirical error bounds, upper-level state value

is the ability of our model to explicitly capture and react to time-varying environment dynamics. As discussed in the "transportation infrastructure" example in Appendix A, even if environment is optimally configured once, its transition dynamics may naturally deteriorate over time if no furhter configuration actions are taken. (For example, roads degrading without maintenance.)

Classical CMDP approached assume that after configuration, the environment's transition kernel changes from $P$ to a better kernel $P'$, and that $P'$ remains fixed during test-time. They treat configuration as a one-shot operation, and do not model how the environment may drift back to suboptimal or deteriorated dynamics.

However, our bi-level formulation is fundamentally different: The upper-level MDP allows the agent to continuously configure the environment over time. This enables the agent to respond to and counteract the natural deterioration of the underlying transition kernel by taking sequential configuration actions.

To empirically compare with the one-shot configuration baseline, we modify the synthetic two-layer example by adding a "no-changing-rate" action to the high-level MDP. If the agent chooses this action, the lower-level kernel naturally deteriorates toward the "bad" environment.

Using value iteration on the upper-level MDP (with four actions: increase rate, decrease rate, keep rate, and no-change-rate), we find that the optimal policy selects one of the first three configuration actions at every upper-level episode $k$.

For the baseline comparison during test-time, the agent is allowed to configure the environment only once at the first time step in classic CMDP baseline. For all subsequent steps, it is forced to take the "no-change-rate" action, making the environment naturally deteriorate to the "bad" kernel. While in our bi-level model, the agent is allowed to continuously execute the optimal upper-level configuration policy at every time step.

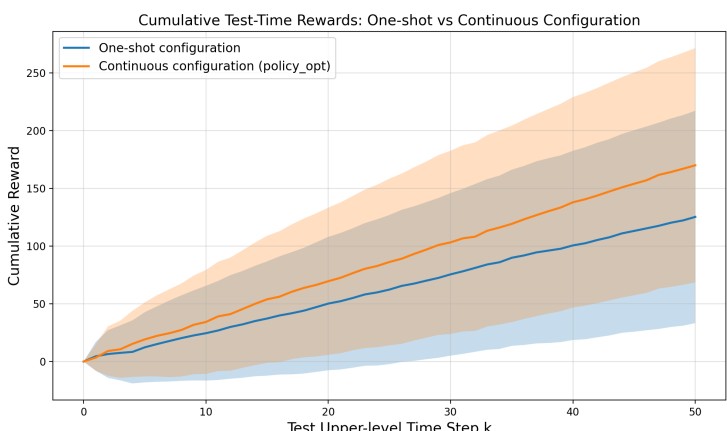

Figure 6: Comparison with baseline one-shot CMDP

The cumulative rewards over 50 episodes during test-time are shown in Figure 6. The shaded region is to indicate the deviations of the test. We can see that our modeling is obviously better than one-shot configuration methods.

### H.3 SENSITIVITY TEST ON THE SYNTHETIC TVCMDP EXPERIMENT

In the synthetic TVCMDP experiment, the cost function on the change of the transition kernel x defined by $C(\mathrm{x}) = \sum_{ij} \beta(e^{\alpha|\mathrm{x}_{ij}|} - 1)$. In Figure 1a, $\alpha = 4$ and $\beta = 1$. We test the sensitivity of the configured state values with respect to $\alpha$ and $\beta$ by varying the values of $\alpha \in [0.5, 8]$ and $\beta \in [0.5, 8]$.

The results show that the configured rewards of our optimization method decrease steadily as the cost function increases with parameters $\alpha$ and $\beta$ separately.

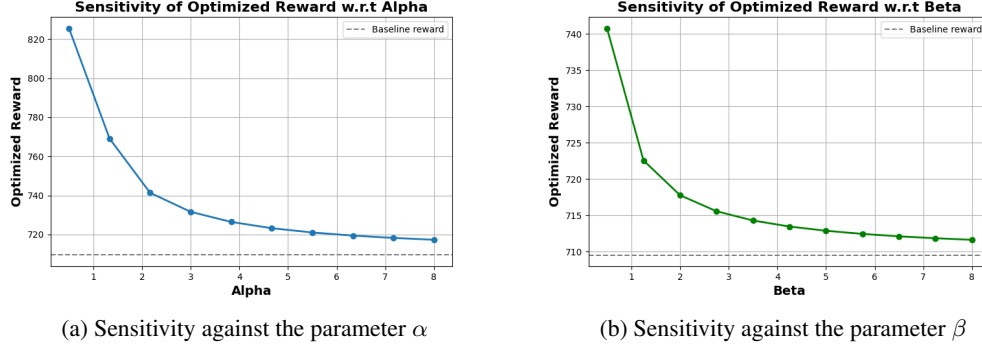

(a) Sensitivity against the parameter $\alpha$         (b) Sensitivity against the parameter $\beta$

Figure 7: Sensitivity of optimization method on TVCMDP

## I  USE OF LLMS

LLMs like ChatGPT are only used for polishing up writing in this paper.

