# OpenReview forum: "Learn to change the world: Multi-level reinforcement learning with model-changing actions"
_ICLR.cc/2026/Conference — Submitted to ICLR 2026_

### Official Review · Reviewer_jxV9 · 2025-10-30

**Soundness:** 2
**Presentation:** 3
**Contribution:** 2
**Rating:** 4
**Confidence:** 3

**Summary:**

The paper formalizes reinforcement learning with model-changing actions—actions that modify the environment’s dynamics—and introduces multi-level configurable MDPs where an upper-level MDP selects (or configures) the lower-level transition kernel, and a special time-variant configurable MDP (TVCMDP) with continuous configuration under a cost budget. Theoretical pieces include a linearization of value w.r.t. transition kernels, a convex surrogate for TVCMDP with exponential configuration costs, and error bounds for bi-level estimation and configuration uncertainty. Algorithms include a model-based bi-level value iteration procedure that alternates solving lower-level MDPs and an upper-level configuration MDP. Experiments on a synthetic TVCMDP, a synthetic bi-level MDP, and two standard tasks—Cartpole and Gridworld/Block-world—demonstrate that learned configuration policies can outperform non-configuring or random-configuring baselines and move performance closer to an “oracle” fixed-best kernel.

**Strengths:**

The paper makes the idea of environment-reconfiguration actionable by cleanly separating configuration (upper level) from control (lower level) and by treating the lower-level transition kernel itself as the state of the upper level—an uncommon but crisp abstraction; it provides a tractable convex approximation for TVCMDP via a linearization of the value function and a budgeted exponential cost, along with bi-level error bounds tying configuration/estimation deviations (δc, δg, Δ) to value degradation; the algorithmic template (solve lower-level values/policies, then plan over kernels) is simple and broadly applicable; and the case studies (Cartpole with discrete parameterizations; Block-world with slip parameter; synthetic TVCMDP/bi-level) collectively show that configuration can materially improve returns over non-configuring or random policies and approach an oracle that fixes the best kernel.

**Weaknesses:**

1. Empirical scope and baselines are limited for claims about generality and efficiency: the upper-level uses value iteration (Cartpole) or DQN (Block-world) over small, discretized configuration spaces, and lower-level policies are DQN or value iteration; there is no comparison to known configurable-MDP solvers or modern bilevel/meta-RL approaches, making it hard to attribute gains to the proposed formulation rather than to problem simplicity.

2. Assumptions in theory are strong and somewhat idealized: known reward functions and the ability to estimate discrete kernel sets for the upper level; while the error bounds are clean, experiments do not probe sensitivity to misspecification or to continuous/large kernel spaces.


3. Evaluation design can blur configuration power with training protocol advantages: for Cartpole the upper level deterministically switches among four handcrafted environments and evaluates over only 20 episodes per setting; Block-world precomputes rewards via offline VI and discretizes $\alpha$ into 1000 points, raising questions about scalability and online sample efficiency when such precomputation is infeasible.


4. Problem framing relative to prior configurable-MDP work could be sharper: while the paper argues that “the lower-level kernel as upper-level state” is novel, the empirical section does not directly compare to CMDP baselines (e.g., gradient or Stackelberg formulations) on the same tasks to evidence distinct benefits of an explicit upper-level MDP abstraction.

**Questions:**

1. For TVCMDP, how sensitive are solutions to $\alpha, \beta$, and budget $B$ in the exponential cost, and where does the linearization break (e.g., $‖x‖$ bounds), empirically?


2. How does performance scale when the configuration state space grows (e.g., $>4$ Cartpole parameter sets or multi-parameter continuous kernels)?


3. Could you report wall-clock, env-step budgets, and seeds for upper- and lower-level training, and provide return-vs-time curves to separate algorithmic from engineering speedups?

4. In the synthetic bi-level study, can you vary $\delta_c$, $\delta_g$, and $\Delta$ to empirically validate the proved error bounds (Lemmas 2–3) and show slopes consistent with theory?

---

> ### Author Response · Authors · 2025-11-26
> **Reply to Questions**
>
> We thank the reviewer for the thoughtful comments. We have addressed all the questions, and we truly appreciate your suggestions on improving our experiments.
>
> 1.  In the synthetic TVCMDP experiment, the cost function on the change of the transition kernel $\mathrm{x}$ is defined by $C(x)=\sum_{ij}\beta (e^{\alpha|\mathrm{x_{ij}}|}-1)$. In Figure 1(a), $\alpha=4$ and $\beta=1$. Figure 1(a) shows that the expected gain increases steadily as the reward $B$ increases.
>
> We test the sensitivity of the configured state values with respect to $\alpha$ and $\beta$ by varying the values of $\alpha\in[0.5,8]$ and $\beta\in[0.5,8]$. The results show that the configured rewards of our optimization method decrease steadily as the cost function increases with parameters $\alpha$ and $\beta$ separately. Please refer to Figure 5 in Appendix H.3 in the revised paper for the sensitivity test.
>
> 2.  To test the performance scale when the configuration state space grows, we add Cartpole parameter sets (upper-level states) from 4 sets to 6 sets and 8 sets separately, with added parameter sets (9.8, 2.0, 0.2, 0.5), (9.8, 2.0, 0.1, 1.0), (9.8, 1.0, 0.2, 1.0), (9.8, 2.0, 0.2, 1.0). It turns out that the average bi-level configured reward gain is increasing as the number of parameter sets increases, from 66.78 (4 upper states), 246.93 (6 upper states), to 246.94 (8 upper states).
>
> 3. The wall clock for the total 8 lower-level models' training is 3,620s. For the upper-level model, since we are doing planning with value iteration, it only takes milliseconds to do so. The total number of env-steps for the 8 lower models is 434,392. The seed for training the DQN network is 42. The increased performance is due to the algorithm instead of the engineering speedups, which can be seen from the comparison of the non-configured results and the configured results in Figure 1(b). All the settings and parameters of both trials are the same, except that the configured result is obtained with the bi-level configuration algorithm.
>
> 4. We thank the reviewer for the comment on empirically verifying the theoretically derived performance bound, and we do think that testifying the error bound numerically should be an important addition to this paper. In the synthetic Bi-level MDP, we separately perturb: 1) the lower-level transition kernel $P(\cdot|s,a),\forall s,a$ using $\delta_g+\delta_c\in[0,0.25]$ and we denote the noisy lower-level kernel with $P'$, and 2) the upper-level model kernel $Q(\cdot|P,b),\forall P,b$ using $\Delta\in[0,0.025]$, and we denote the noisy upper-level kernel with $Q'$. This allows us to isolate how lower-level and upper-level errors influence the performance of  Algorithm 1
> Figure 4 in Appendix H.2 reports the resulting lower-level state value errors between $P$ and $P'$. The blue curve shows the infinity norm difference between the true and perturbed lower-level value functions. We sample noisy lower-level kernels $P'$ from a distribution of noisy kernels such that $TV(P,P')=\delta_g+\delta_c$ and record the maximum difference between state values (namely maximum $\ell_{\infty}$ norm of difference) among all the realizations of the perturbations. The error grows linearly with the perturbation size and stays within the theoretical bound.
>
> Figure 5 in Appendix H.2 reports the resulting upper-level state value errors separately by perturbing the lower-level kernel and the upper-level kernel. In the first case, $\Delta=0$, and in the second case, $\delta_g+\delta_c=0$. The blue curves show the infinity norm difference between the true and perturbed upper-level value functions. The upper-level state value error grows linearly with the two perturbation sizes separately and stays within the theoretical bound stated in Lemma 3.

---

> ### Author Response · Authors · 2025-11-26
> **Reply to Weakness**
>
> Comparison with classic configurable MDPs baselines:
>
> The key distinction between our bi-level configurable MDP and the classical CMDP formulations, such as gradient CMDP and Stackelberg CDMP, is the ability of our model to explicitly capture and react to time-varying environment dynamics.  As discussed in the ``transportation infrastructure'' example in Appendix A, even if the environment is optimally configured once, its transition dynamics may naturally deteriorate over time if no further configuration actions are taken. (For example, roads are degrading without maintenance.)
>
> Classical CMDP approaches assume that after configuration, the environment's transition kernel changes from $P$ to a better kernel $P'$, and that $P'$ remains fixed during test-time. They treat configuration as a one-shot operation and do not model how the environment may drift back to suboptimal or deteriorated configurations.
>
> However, our bi-level formulation is fundamentally different: The upper-level MDP allows the agent to continuously configure the environment over time. This enables the agent to respond to and counteract the natural deterioration of the underlying transition kernel by taking sequential configuration actions.
>
> To empirically compare with the one-shot configuration baseline, we modify the synthetic two-layer example by adding a "no-changing-rate" action to the high-level MDP. If the agent chooses this action, the lower-level kernel naturally deteriorates toward the "bad" environment.
>
> Using value iteration on the upper-level MDP (with four actions: increase rate, decrease rate, keep rate, and no-change-rate), we find that the optimal policy selects one of the first three configuration actions at every upper-level episode $k$.
>
> For the baseline comparison during test-time, the agent is allowed to configure the environment only once at the first time step in the classic CMDP baseline. For all subsequent steps, it is forced to take the ``no-change-rate'' action, making the environment naturally deteriorate to the "bad" kernel. While in our bi-level model, the agent is allowed to continuously execute the optimal upper-level configuration policy at every time step.
>
> The cumulative rewards over $50$ episodes during test-time are shown in Figure 6 of Appendix H.2. The shaded region indicates the deviations of the test. We can see that our modeling is obviously better than one-shot configuration methods.

---

### Official Review · Reviewer_7Tap · 2025-10-31

**Soundness:** 3
**Presentation:** 3
**Contribution:** 3
**Rating:** 8
**Confidence:** 5

**Summary:**

The paper introduces the multi-layer configurable time-varying Markov decision process (MCTVMDP) with model-changing actions that actively modify the RL model of world dynamics itself. The new RL framework proposed in this paper has the potential and mechanism of breaking out of the current model, through actions that change or improve the underlying MDP. The authors also study special cases of the proposed RL mechanism with model-changing actions; they consider configurable RL for time-varying environments and multi-level (including bi-level) environment-changing RLs. They propose a multi-level value iteration algorithm for solving multi-level configurable RL problems. Lastly, there are some experiments done in Cartpole and Block-world environments to conclude that the proposed framework is adaptable to more complex and continuous RL environments and to testify to the feasibility of the bilevel configurable MDP model.

**Strengths:**

Although the notion of configurable MDP (CMDP) has previously been proposed, the main strengths can be summarized as follows:
1. The primary contribution of this research lies in the study of multi-layer configurable MDPs.
2. The authors employ upper-level MDP abstractions/learning to model/improve configuration actions.
3. Additionally, they deal with time-varying non-stationary lower-level MDPs (time-varying transition kernels), which were not investigated by these previous works.
4. They introduce a model-based value iteration algorithm.

**Weaknesses:**

Some questions and concerns must be addressed for both theory and experiments.
1. How do the upper-level model and lower-level model connect?
2. On what basis does the agent select actions?
3. In the Bi-level model-based value iteration algorithm, how much time does it take to converge to the optimal policy?
4. How would your bi-level model-based value iteration algorithm’s performance change with different environments and different algorithms other than the DQN?
5. Why did you specifically choose to employ the DQN algorithm?
On page 18,  figure 3, why is the performance of Oracle (blue bar) closest to the performance of the bi-level MDP configuration?

Minor comments:
1. On page 9, “We present the configurable Carpole”, should be Cartpole.
2. On page 5, section 3, line 1: Should be TVCMDP instead of TVCDMP.

**Questions:**

I would highly recommend including a conclusion that summarizes the main notion and contributions.

---

> ### Author Response · Authors · 2025-11-26
> **Reply to Weakness**
>
> We thank the reviewer for your effort of commenting and reviewing this paper. We really appreciate your remark on the novelty of this paper.
>
> 1. In our bi-level configurable MDP, the upper-level and lower-level models are connected through the transition kernel of the lower-level MDP. The upper-level state space $\mathcal{P}$ is composed of all possible lower-level transition kernels. Thus, each upper-level state $P\in\mathcal{P}$ represents a specific instance of the lower-level environment.
>
> At each episode $k$, the agent selects a model-changing action $b_k=\Theta(P_{k-1})$, where $\Theta:\mathcal{P}\rightarrow \mathcal{B}$ is the configuration policy. The action determines how the transition kernel of the lower-level model is updated, according to the upper-level model transition dynamics:  $P_k\sim Q(P_k|P_{k-1}, b_k)$.
>
> Once the new kernel $P_k$ is set, the lower-level MDP runs under this transition model and produces an expected return $J(\pi^*_k, P_k)$. The upper-level reward function uses this lower-level performance to define the reward of the upper-level decision.
>
> So, in a nutshell, the connections are: First, the upper-level action $b_k$ chooses the transition kernel for the lower-level MDP. Second, the lower-level performance feeds back as the reward to the upper-level model.
>
> 2. The agent selects action at two levels, each governed by a different objective. At the upper-level $\mathcal{M}_U$, the agent chooses the model-changing action $b$ according to a deterministic configuration policy $\Theta$. At the lower-level $\mathcal{M}_L$, it chooses primitive actions $a$ according to a deterministic control policy $\pi$.
>
> The basis for these choices is as follows: The control policy $\pi_k$ of episode $k$ is chosen to maximize the lower-level expected return $V^{\pi_k}$ under the current transition kernel $P_k$. The configuration policy $\Theta$ is selected to maximize the upper-level expected return $W^{\Theta}$, which depends on the lower-level performance across episodes.
>
> 3. We thank the reviewer for raising this question. Convergence time is indeed a key aspect of evaluating the performance of our bi-level value iteration algorithm. Empirically, in our synthetic example, the bi-level value iteration algorithm converges to the optimal policy in a total of 1007 iterations, with three lower-level environments converging separately with 252, 225, 220 iterations, and the upper-level converging with 307 iterations.
>
> Analytically, the convergence time can be understood by extending the standard value-iteration convergence analysis. Recall that for a one-layer (standard) MDP, the number of iterations $n$ needed for value iteration to reach an $\epsilon$-accurate estimate satisfies: $n\geq \log(2\|V_1 - V_0\|_{\infty}/\epsilon(1-\gamma))/\log(1/\gamma)$, where $V_0$ is the initial value estimate, $V_1$ is the estimate after one Bellman update, and $\epsilon$ is the accuracy gap.
>
> In the bi-level case, suppose the upper-level state space contains $d$ possible lower-level models. For each model $P_i\in\mathcal{P}$, let $n_i$ denote the number of iterations required for the lower-level value iteration to cover to its $\epsilon$-optimal value. Then the total number of lower-level iterations needed to compute all lower-level value functions is $n^*=\sum_{i=1}^dn_i$.
>
> Once these lower-level values are computed, the upper-level value iteration proceeds on the $d$-state upper-level MDP. Using an argument analogous to the one-layer case, the upper-level convergence time can be expressed in terms of a similar contract-based analysis; We denote this number of iterations by $N$. A full theoretical derivation of $N$ is an interesting direction for our future work, but it is entirely feasible.
>
> Thus, the total number of iterations required for the bi-level value iteration to converge to the optimal policy is $N+\sum_{i=1}^d n_i$.
>
> 4. The performance of the bi-level model remains steady with algorithms other than DQN and with different environments. Please see our supplemental experiments in Appendix H.2 ``Comparison with classic configurable MDPs'' in the revised paper.

---

> ### Author Response · Authors · 2025-11-26
> **Reply to Weakness (Continued)**
>
> 5. We used the DQN algorithm in our experiments because it is a value-based method, which allows us to directly access state-action values. This aligns naturally with our bi-level value iteration algorithm, which is also value-based: When DQN is used at the lower-level, its learned value estimates can be reused to efficiently compute the upper-level reward function.
>
> In practice, however, the lower-level RL algorithm does not have to be DQN. If we relax the upper-level reward function so that it does not require explicit lower-level state values, for example, the empirical performance the lower-level agent, then any RL method can be used at the lower level. In that case, the upper-level model would simply treat the lower-level algorithm as a black-box whose performance can be evaluated after each episode.
>
> Similarly, in the block-world example, we chose DQN  to optimize the upper-level model parameter $\alpha$ because it handles continuous states well, but this choice is not restrictive. Any RL method that can handle the continuous state can be a substitute.
>
> In Figure 3, page 18, the Oracle represents the maximum achievable reward when the agent is always placed in the best possible lower-level environment and uses the optimal primitive policy. The performance of the bi-level MDP configuration approaches this oracle because bi-level value iteration discovers the optimal configuration policy that continually improves the lower-level environment across episodes. In contrast, the "random configuration" baseline samples environments arbitrarily, and the "no-configuration" baseline never adapts the environments at all.
>
> Minor issues: We thank the reviewer for your careful catch! We have fixed those typos. We have added a conclusion section in the revised paper, too.

---

### Official Review · Reviewer_P38m · 2025-11-01

**Soundness:** 1
**Presentation:** 2
**Contribution:** 2
**Rating:** 2
**Confidence:** 3

**Summary:**

The paper introduces a MDP "inception" framework: a two-level process where the upper-level MDP action picks a configuration that affects the transition kernel of the lower-level MDP. Rewards of the upper level MDP are defined as the expected return of the next episode of the lower-level optimal policy under the chosen configuration after substracting a cost for "changing the world": the configuration is picked by perturbing previous transition kernel and the cost function reflects the price of the perturbation (chosen as exponential function).
The authors also provide a first-order value linearization to turn continuous kernel tweaks into a convex program, as well as estimation and propagation error bounds (based on classical results from RL). The experiments use the famous cartpole environment as toy example to validate the theoretical claims.
The related work is addressed, especially Configurable Markov Decision Processes, which is probably the most relevant. "Contextual Bilevel Reinforcement Learning for Incentive Alignment" also presents similar ideas, but in the case where the "world changes" are not controlled by the upper-level MDP.

**Strengths:**

The authors claim novelty relative to configurable RL by targeting time-varying, non-stationary RL and continuous configuration. While prior work (as stated in the summary) also addresses similar settings, the lower–upper decoupling perhaps allows a clearer formulation of estimation and error bounds by leveraging well-established results from tabular RL. The writing is clear and easy to follow. The main idea can be understood even by readers with limited prior exposure to reinforcement learning.

**Weaknesses:**

While the proposed two-level formulation offers an appealing perspective and an intuitive conceptual separation, the same setting can be expressed within the standard Markov Decision Process (MDP) framework: this can be achieved by augmenting the state and action spaces such that the action space becomes the Cartesian product of the configuration and the lower-level action spaces. In such formulation, the configuration component is selected once at the beginning of each episode, while the lower-level actions are kept as is. The state space remains unchanged, but the configuration affects the transition dynamics; consequently, the overall process can be represented by a single transition kernel conditioned on the joint action pair (configuration, action). This perspective implies that the proposed hierarchical structure may be viewed as a reformulation of an augmented flat MDP, rather than a fundamentally new class of decision process.

We also raise concerns regarding the level of academic rigor in the paper’s presentation. The introduction, while comprehensive, often reads as overly verbose and could benefit from greater synthesis. For example, the second and third paragraphs substantially overlap in content and could be merged, and the sporadic use of quotation marks around common reinforcement learning terminology (e.g., “MDP,” “infinity,” “infinite-horizon”) yields a conversational tone, which should be avoided in a conference submission. Adopting a more precise and concise tone throughout would significantly enhance the professionalism and readability of the paper.

While the CartPole simulations are not sufficient indicators, we believe that the main flaw of this paper lies in the optimization objective:
- The authors used a linearization $V_\pi(P^\pi + x) \approx N + MxN$, with $M = \gamma (I - \gamma P_\pi)^{-1}$ and $N = (I - \gamma P_\pi)^{-1} r_\pi$ for a fixed $\pi$, so a more accurate notation would be $N^\pi + M^\pi x N^\pi$.
- Accordingly, the objective becomes $\max_{x_k} \max_{\pi} N^\pi + M^\pi x N^\pi$, and $N^\pi$ can no longer be omitted, contrary to the claim in the paper.
- Moreover, the objective is no longer convex, at least in the general case, as it is not shown to be jointly convex in both $x$ and $\pi$. A simple example is the function $(x, y) \mapsto xy$, which is convex in each variable individually, but not jointly convex.
- Even if we disregard this issue (which by itself undermines the proposed approach), the linearization is only valid in a neighborhood of the current $P^\pi$, where $|x_{ij}| \leq \max P^\pi_{i,j}$ (not merely $|x_{ij}| \leq 1$). Consequently, if the proposed scheme is to be followed, the modification $x$ of the kernel must be bounded at each step rather than freely vary in the $[-1, 1]$ interval, thereby requiring a "trust-region" optimization formulation.
- The cost bound does not seem to be used in Algorithm1 nor in its low-level formulation.

**Questions:**

The main question at this point concerns the linearization. We expect a solid and rigorous proof of any proposed modification to salvage the paper. Although we recognize that such an effort may be substantial within the given time frame, we are willing to raise our rating if it is adequately addressed.

---

> ### Author Response · Authors · 2025-11-22
> **Reply to Questions**
>
> We appreciate the efforts of reviewer's for giving a thoughtful and careful review.
>
> Main questions regarding linearization:
>
> $\cdot$ We agree with the reviewer that the notation for $N$ and $M$ should be replaced by $N^{\pi}$ and $M^{\pi}$, since these two variables depend on the policy $\pi$. We have modified all $N$ to $N^{\pi}$, and all $M$ to $M^{\pi}$ in the revised paper.
>
> $\cdot$ We omit $N^{\pi}$ from the objective function $\max_{\pi}\max_x N^{\pi}+M^{\pi}xN^{\pi}$ based on the assumption that the optimal policy $\pi$ does not change if a small enough change in the transition kernel is made. Even if the assumption (that optimal policy does not change)  does not hold, keeping the policy unchanged provides a meaningful lower bound of the objective function, compared with optimizing the policy at the same time.
>
> In line 256 of the original paper, we assume that the change $x$ in the transition kernel is small enough that the optimal policy corresponding to the configured kernel $P+x$ is still $\pi$, so $\max_{\pi}$ serves as a ``dummy operator'' here. We rewrite the this assumption to make it more clear.
>
> Note that by fixing the policy $\pi$, $\max_x N^{\pi}+M^{\pi}xN^{\pi}$ can serve as a lower bound on achievable V-values (since we have not optimized the policy yet after changing kernel $P$). Our method works like this: for the current unchanged transition kernel, we find the optimal policy $\pi$. Then we fix the policy $\pi$ and use the linearization to maximize this linearized lower bound via optimizing over $P$. When we get to a new $P$, we can then optimize for a possibly new optimal policy $\pi$ for this new kernel $P$. We can iterate this process if needed. In this iterative process, we do not have to keep the $N^{\pi}$ in the optimization process of each linearization, since we are fixing $\pi$ when we do that linearized optimization.
>
> $\cdot$ In our method, we did not use the joint convexity of $\pi$ and $x$, and we only used convexity in $x$ since we fixed $\pi$ when we do linearization.  However,  we remark that, even if we jointly optimize $\pi$ and $x$, the optimization objective function $f(x)=\max_{\pi}N^{\pi} + M^{\pi}xN^{\pi}$ is concave with respect to the single variable $x$, when we take into considerations maximization over different policies.  Note that  for each fixed $\pi$, the objective function $N^{\pi} + M^{\pi}xN^{\pi}$ is concave (affine) in $x$. So  $\max_{\pi}N^{\pi} + M^{\pi}xN^{\pi}$ is also a concave function over $x$, because  the maximum (over different policies) of concave functions is also a concave function. So if we consider a optimization problem maximizing $f(x)$, it is still a convex optimization problem (the negative of the objective function is a convex function) in $x$, even if we incorporate $\max_{\pi}$ into the objective function.
>
> $\cdot$ We agree with the reviewer, the linearization is indeed valid only within a small neighborhood of $P^{\pi}$.  We agree that we should have added the constraints $|x_{ij}|\leq \max P^{\pi}_{ij}$ \emph{in writing} to the optimization problem. We have added this (modified) constraint back to the optimization problem, and obtained a new set of numerical results (whcih are similar to earlier results). In addition, for the reason mentioned below, we were already doing a trust-region optimization,  implicitly forcing the feasible region to be small.
>
>  We would like to mention that configuring $x$ should be very costly due to physical constraints. To assure that, we placed constraint on the variable $x$ as $\sum_{k,i,j}(e^{\alpha |(x_k)_{ij}|}-1)\leq B$, and in our code implementations, the constant $\alpha$ is large. So essentially the optimization will return a solution with a small magnitude for the change.  This also makes sure the change is within small neighborhood of the original $P$, and the corresponding optimal policy likely does not change within a small enough in $P$ (even if the optimal policy changes, the objective function still provides a meaningful lower bound compared with optimizing policy at the same time).
>
> $\cdot$ We think there is a misunderstanding. Algorithm 1 is for solving  the general Bi-level configurable MDP, not for solving the optimization problem for the time-variant configurable MDP (TVCMDP), for which the optimization formulation (6) instead works. That is why we did not use (6) in Algorithm 1. In the general Bi-level configurable MDP, the model-changing actions $b$ change the lower-level environment through a MDP process and the cost of model-changing actions is in the form of the penalty term  $-C(P_k,b_k)$ in the reward function of the higher-level MDP.

---

> ### Author Response · Authors · 2025-11-22
> **Reply to Weakness**
>
> 1. Linearization: please see reply to "Questions".
>
> 2. Difference between two-level formulation and standard large MDP: Respectfully, the idea the reviewer suggests does not work as intended for our problem and on the contrary proves our formulation in this paper is new. In addition, the suggestion likely only considered the special case TVCMDP (time variant configurable MDP) rather than the general multi-layer MDPs with model-changing actions (of which TVCMDP is only a special case).
>
> According to the reviewer's suggestion, one needs to essentially enlarge the action space to include both configurations and lower-level actions. The suggestion states that ``the state space remains unchanged'', and configuration actions allow the transition dynamics to change. However, in this case, these actions will change the transition kernel of the involved MDP: then these are not regular MDP models where transition kernels stay fixed and actions will not affect the transition kernels. These new scenarios are about model-changing actions, which are exactly the proposed innovations in the paper.
>
> Also, the augmented MDP suggested by the reviewer does not involve the dynamics of learning (since it only takes configuration actions at certain time indices, there is no MDP dynamics transitioning between different configurations) in the upper-level MDP which models the dynamic processes of configuring lower-level transition kernels; while our new multi-layer configurable MDP models naturally includes such dynamics.  The paper's ideas naturally extend to $>2$  layers of configurable MDP, but the reviewer's suggested large MDP only works for special cases of $\leq$ 2-layer configurable MDP (namely TCVMDP considered in part of our paper, a special case of 2-level MDP with model-changing actions) where configurable actions work only independently at fixed time indices and there is no MDP model for the dynamics of configuring environments or other parameters.
>
> As another evidence for validating our formulation being a new model, we can look at the literature of configurable MDPs. If we can simply enlarge the action space and treat these configurable MDPs as regular augmented flat MDPs, then there is not much reason for studying configurable MDPs and there is not much point for having the configurable MDP literature. However, it is NOT the case that people simply reduce configurable MDP to regular MDP, and researchers generally agree to treat configurable MDPs as a new research area. This is because configuration actions are different from actions of traditional MDP models, bringing new research challenges.
>
>  On a related but different note from what the reviewer suggests, we would like to discuss another idea of making a regular MDP for this scheme. Even though, in principle, we can transform this scheme (and almost EVERYTHING in the world) to a bigger enough regular MDP, that would require incorporating the lower-level MDP's transition kernels as extra-dimension components of the state representation. This makes the size of the state space much larger, especially if the original underlying MDP itself already has a very large state space. Sometimes, directly optimizing the policies over this enlarged state space may be inefficient, due to not utilizing the distinctions between model-changing actions and non-model-changing actions. Moreover, because the model-changing actions may be permissible not at every time step but only at certain given time steps, one cannot change the scheme of RL with model-changing actions into a regular RL with a fixed set of actions.
>
> Another reason for the new formulation in this paper was also mentioned by the reviewer, ``the proposed two-level formulation offers an appealing perspective and an intuitive conceptual separation'' than simply making a super big regular MDP out of it. The new formulation has an advantage in concise representations.
>
> 3. Introduction Improvement: We thank the reviewer for this comment. The objectives of the second and the third paragraphs in the introduction are, in fact, different. The second paragraph aims to highlight the difference between traditional MDP and the configurable MDP framework with model-changing actions proposed in this work. In contrast, the third paragraph provides a concise overview of the two specific models analyzed in detail later: the time-varying configurable MDP and the bi-level configurable MDP. However, we agree with the reviewer that the second paragraph could be shortened, since the core disparity between traditional MDPs and configurable MDPs has already been discussed in previous literature. We have rewritten these two paragraphs to make them clearer in the revised paper.
>
> We thank the reviewer for pointing out the use of quotation marks. We have deleted those quotation marks on terms like MDP, infinite, and infinite-horizon in the revised version of the paper, and change them to more rigorous writings.

---

### Author Response · Authors · 2025-12-02
**Summary Comments from Authors**

We would like to thank the reviewers, area and program chairs for their hard work in reviewing our paper and helping improve its quality. Reviewers generally praised the novelty of our framework: we formulate a fresh multi-layer MDP system in which the agents do not merely adapt to a fixed environment like in almost all existing RL models, but instead can take novel model-changing actions to select or modify the MDP model itself including modifying its environment. This will enable higher achievable rewards, and can also lead to more advanced RL autonomous systems (for example, the crossing-river robot in our introduction). Reviewers also noted our strong theoretical guarantees, bi-level value-iteration algorithms, and tractable convex approximation for time-varying MDPs.  Reviewer 7Tap gave a high rating (8), quoting from this reviewer: “employ upper-level MDP abstractions/learning to model/improve configuration actions”, and “they deal with time-varying non-stationary lower-level MDPs (time-varying transition kernels), which were not investigated by these previous works” Reviewer P38m indicated willingness to raise their score depending clarification of the linearization, quoting from the reviewer, “The main question at this point concerns the linearization. We are willing to raise our rating if it is adequately addressed.” and Reviewer jxV9’s raised questions regarding the empirical comparison of prior configurable-MDP algorithms, and the empirical verification of the theoretical bounds. We believe that we have fully addressed these concerns.

Specifically, we have improved our original paper in the following aspects, as suggested by the reviewers:

1. Presentation: We have strengthened our introduction, including clarifying the distinction between multi-layer configurable MDPs and classic configurable MDP settings. We added a conclusion section to highlight our contribution to configurable RL and future directions of this work, as suggested by the reviewer.
2. Optimization problem clarification: Reviewer jxV9’s main concern was on the linearization of the optimization problem, but actually, our formulation is solid ( after presentation clarifications). The confusion was about the non-convexity of optimizing the state value function with respect to both the kernel $x$ and the policy $\pi$. We think the confusion was due to the notation $\max_{\pi}\max_{x}$. In fact, in the original version, we have stated the assumption that the kernel change $x$ is small enough such that the policy $\pi$ remains unchanged, and the convexity is only in variable $x$.  The notation $\max_{\pi}$ was to indicate that the agent is looking for the optimal policy before configuration happens. We clarified the assumption for the convex optimization problem to hold: we explicitly stated that kernel changes are small enough so that the optimizing policy does not change. Even if the optimal policy changes, fixing the policy still gives a meaningful lower bound of the improved reward. To make this clear,  we get rid of the notation $\max_{\pi}$ in the optimization problem statement in (6). We also added the constraint that the linearization is valid in the neighbourhood of the original transition kernel, as suggested by the reviewer. We further explained why our new MDP models with model-changing actions cannot be implemented by a regular MDP with enlarged action space as claimed by the reviewer, further clarifying the novelty of our layered MDP with model-changing actions.

3. Empirical enhancement: We have added new experiments comparing the one-time configuration approaches (e.g., gradient-based or Stachelberg-game-based) and our sequential bi-level value-iteration algorithm. We also empirically computed the error bounds under both configuration-induced and empirical noise, confirming consistency with Lemma 3. Finally, we assessed the robustness of our proposed time-varying configurable MDP solver and found it stable across different constraint parameters, as suggested by Reviewer jxV9.

---

### Meta-Review · Area_Chair_cUBx · 2026-01-12

**Summary:**

This paper proposes RL in a multi-layer configurable time-varying MDP with a top-level MDP which selects actions that change configurations, and a set of bottom-level MDPs which have stationary transition policies. Optimization is performed jointly over these two parts.

Main concerns raised were that a standard MDP formalism that express the proposed MCTVMDP and limited nature of empirical evidence. I believe these concerns remain despite author rebuttal. The experiments are on simple domains, and comparison with function-approximation approaches such as PPO would have been nice. Importantly, in these more complex settings, the practical use of MDP vs MCTVMDP is less clear to me, since one can easily append time to state features in the policy to learn to optimize wrt time-varying MDP. I encourage authors to add more experiments on complex domains, and particularly try to show difference between MDP and MCTVMDP.

For these reasons,  I am going with weak reject.

**Reviewer Concerns:**

1. Reviewer P38m raised concerns on lack of mathematical expressivity between MDP and MCTVMDP, and issue with the mathematical formalism. Authors have fixed the concern by adding an extra constraint. Their argument on the expressivity is that it leads to a bigger state space. However, this is not a problem for function-approximation approaches which is the more widely used approach. Therefore, I am not clear about the practical value of the proposed formalism.

2. Reviewer 7Tap didnt raise any major concern, but also didnt provide a strong justification for their score.

3. Reviewer jxV9 raised concerns about limited nature of empirical evidence, idealized theoretical settings (e.g., known reward function), and comparison with respect to previous configurable-MDP work. Authors have added results with more hyperparameters and clarified difference with previous work.

**Reviewer Scores:**

1. Reviewer P38m could have raised their score to 4.

2. Reviewer 7Tap would likely have kept their score at 8.

3. Reviewer jxV9 may have raised their score to 5/6.

---

### Decision · Program_Chairs · 2026-01-26

Reject